# Necessary and Sufficient Conditions for Optimal Decision Trees using Dynamic Programming

**Jacobus G. M. van der Linden**      **Mathijs M. de Weerdt**      **Emir Demirović**
Delft University of Technology, Department of Computer Science
`{J.G.M.vanderLinden, M.M.deWeerdt, E.Demirovic}@tudelft.nl`

## Abstract

Global optimization of decision trees has shown to be promising in terms of accuracy, size, and consequently human comprehensibility. However, many of the methods used rely on general-purpose solvers for which scalability remains an issue. Dynamic programming methods have been shown to scale much better because they exploit the tree structure by solving subtrees as independent subproblems. However, this only works when an objective can be optimized separately for subtrees. We explore this relationship in detail and show the necessary and sufficient conditions for such separability and generalize previous dynamic programming approaches into a framework that can optimize any combination of separable objectives and constraints. Experiments on five application domains show the general applicability of this framework, while outperforming the scalability of general-purpose solvers by a large margin.

## 1   Introduction

Many high-stake domains, such as medical diagnosis, parole decisions, housing appointments, and hiring procedures, require human-comprehensible machine learning (ML) models to ensure transparency, safety, and reliability [74]. Decision trees offer such human comprehensibility, provided the trees are small [71]. This motivates the search for *optimal* decision trees, i.e., trees that globally optimize an objective for a given maximum size.

The aforementioned application domains wildly differ from one another, and therefore the objective and constraints also differ, leading to a variety of tasks: e.g., cost-sensitive classification for medical diagnosis [32, 82]; prescriptive policy generation for revenue maximization [13]; or ensuring fairness constraints are met [1]. For broad-scale application of optimal decision trees, we need methods that can *generalize* to incorporate objectives and constraints such as the ones from these application domains, while remaining *scalable* to real-world problem sizes.

However, many state-of-the-art methods for optimal decision trees lack the required scalability, for example, to optimize datasets with more than several thousands of instances or to find optimal trees beyond depth three. This includes approaches such as mixed-integer programming (MIP) [10, 86], constraint programming (CP) [84], boolean satisfiability (SAT) [42] or maximum satisfiability (MaxSAT) [34].

Dynamic programming (DP) approaches show scalability that is orders of magnitude better [3, 20, 52] by directly exploiting the tree structure. Each problem is solved by a recursive step that involves two subproblems, each just half the size of the original problem. Additional techniques such as bounding and caching are used to further enhance performance.

However, unlike general-purpose solvers, such as MIP, DP cannot trivially adapt to the variety of objectives and constraints mentioned before. For DP to work efficiently, it must be possible to solve the optimization task at hand *separately* for every subtree.

37th Conference on Neural Information Processing Systems (NeurIPS 2023).

Therefore, the main research question considered here is to what extent can this separability property be generalized? In answering this question we provide a generic framework called *STreeD* (Separable Trees with Dynamic programming). We push the limits of DP for optimal decision trees by providing conditions for separability that are both necessary and sufficient. These conditions are less strict and extend to a larger class of optimization tasks than those of state-of-the-art DP frameworks [52, 66]. We also show that any combination of separable optimization tasks is also separable. We thus attain generalizability similar to general-purpose solvers, while preserving the scalability of DP methods.

In our experiments, we demonstrate the flexibility of STreeD on a variety of optimization tasks, including cost-sensitive classification, prescriptive policy generation, nonlinear classification metrics, and group fairness.

As is commonly done, we assume that features are binarized beforehand. Therefore, STreeD returns optimal *binary* decision trees under the assumption of a given binarization.

In summary, our main contributions are 1) a generalized DP framework (STreeD) for optimal decision trees that can optimize any *separable* objective or constraint; 2) a proof for necessary and sufficient conditions for separability; and 3) extensive experiments on five application domains that show the flexibility of STreeD, while performing on par or better than the state of the art.

The following sections introduce related work and preliminaries. We then further define separability, show conditions for separability, and provide a framework that can find optimal decision trees for any separable optimization task. Finally, we provide five diverse example applications and test the performance of the new framework on these applications versus the state-of-the-art.

## 2   Related Work

**Decision tree learning**   Because optimal decision tree search is an NP-Hard problem [38], most traditional solution methods for decision tree learning were heuristics. These heuristics, such as CART [14] and C4.5 [73], often employ a top-down compilation, splitting the tree based on a local information gain or entropy metric. However, for each new objective, new splitting criteria need to be devised, and often this splitting criterion only indirectly relates to the real objective.

With increasing computational power, optimal decision tree search for limited depth has become feasible. Bertsimas and Dunn [10] and Verwer and Zhang [85] showed that optimal trees better capture the information of the dataset than heuristics by directly optimizing the objective at hand, resulting on average in a 1-5% gain in out-of-sample accuracy for trees of similar or smaller size.

However, their MIP models –and also the state-of-the-art MIP models [2, 37, 86, 93]– may often take several hours for medium-sized datasets, even when considering small trees up to depth three. Others have proposed to use SAT [8, 42, 64], MaxSAT [34, 76], and CP [84]. These methods also lack scalability [20], or –in the case of SAT– solve a different problem: finding a minimal tree with no misclassifications. Because scalability is such an issue for these methods, many propose to solve the problem with variants of local search, such as coordinate descent [11, 26], tree alternating optimization [15] and a heuristic SAT approach [75].

In contrast, DP approaches find globally optimal trees while achieving orders of magnitude better scalability [3], so the survey by Costa and Pedreira [18] concludes that specialized DP methods are the most promising future direction for optimal decision trees. Nijssen and Fromont [65] proposed DL8, the first dynamic programming approach for optimal decision trees, which they later generalized to a broader set of objectives and constraints [66]. Aglin et al. [3, 4] improved the scalability of DL8 by adding branch and bound and new caching techniques to the updated algorithm DL8.5. Hu et al. [36] and Lin et al. [52] added sparsity-based pruning and new lower bound techniques. Demirović et al. [20] improved scalability by adding a special solver for trees of depth two, a similarity lower bound, and other techniques. Recent developments include an anytime algorithm [21, 67] and learning under memory constraints [66].

**Objectives and constraints**   Decision tree learning can be extended by considering a variety of constraints. Nanfack et al. [63] categorized these constraints into structure, attribute, and instance constraints. Structure constraints limit the structure of the tree, such as its depth or size. Attribute constraints impose limits based on the attribute values. Examples are monotonicity constraints [35], cost-sensitive classification [55] and group fairness constraints [44]. Instance constraints operate on

pairs of instances, such as robustness constraints [88], individual fairness constraints [1], and must- or cannot-link constraints in clustering [78]. Among these, structure constraints are most widely included in decision tree learning methods.

Many of these constraints or objectives can be added to MIP models. Therefore, the literature shows a multitude of MIP models for a variety of applications, such as fairness [1], algorithm selection [87], and robustness [88]. However, scalability remains an issue.

Nijssen and Fromont [66] show how the DP-based DL8 algorithm [65] can be generalized for a variety of objectives, provided that the objectives are *additive*, i.e., the value of an objective in a tree node is equal to the sum of the value in its children. Constraints are required to be *anti-monotonic*, i.e., a constraint should always be violated for a tree whenever it is violated for any of its subtrees. They consider constraints and objectives such as minimum support in a leaf node, C4.5's estimated error rate, Bayesian probability estimation, cost-sensitive classification, and privacy preservation.

Since then, new DP approaches have improved on the runtime performance of DL8. This, however, reduced the generalizability of those methods. DL8.5 [3, 4], for example, can optimize any additive objective but does not support constraints. MurTree [20] provides new algorithmic techniques that result in much better scalability but only considers accuracy. GOSDT [52] can optimize additive objectives that are linear in the number of false positives and negatives but also does not support constraints. When the objective is nonlinear, they no longer use DP. Demirović and Stuckey [19] show how DP can be used to optimize even nonlinear functions, provided the objective is monotonically increasing in terms of the false positives and negatives, but they also do not consider constraints.

**Summary**    Decision tree learning is often approached through top-down induction heuristics, which depend on developing complex custom splitting criteria for each new learning task and may perform suboptimally in terms of classification accuracy. Optimal methods often outperform heuristics in accuracy but typically lack scalability. Optimal DP approaches have better scalability but lack generalizability to other objectives and constraints. In the next sections, we address this by pushing the limits of what can be solved by a general yet efficient DP framework for optimal decision trees.

## 3    Preliminaries

This section introduces notation, defines the problem, and explains how dynamic programming solutions for optimal decision trees work.

**Notation and problem definition**    Let $\mathcal{F}$ be a set of features and let $\mathcal{K}$ be a set of labels that are used to describe an instance. Let $\mathcal{D}$ be a dataset consisting of instances $(x, k)$, with $x \in \{0, 1\}^{|\mathcal{F}|}$ the feature vector and $k \in \mathcal{K}$ the label of an instance. Because the features are binary, we introduce the notation $f$ and $\bar{f}$ for every feature $f \in \mathcal{F}$ to denote whether an instance satisfies feature $f$ or not. In the same way, let $\mathcal{D}_f$ describe the set of instances in $\mathcal{D}$ that satisfy feature $f$, and $\mathcal{D}_{\bar{f}}$ the set of instances that does not.

Let $\tau = (B, L, b, l)$ be a binary tree with $B$ the internal branching nodes, $L$ the leaf nodes, $b : B \to \mathcal{F}$ the assignment of features to branching nodes and $l : L \to \mathcal{K}$ the assignment of labels to leaf nodes. The left and right child nodes of a branching node $u \in B$ are given by $u_L$ and $u_R$, respectively. Instances that satisfy a feature branching test are sent to the right subtree, and the rest to the left.

Then, for example, when minimizing misclassification score (the number of misclassified instances), the cost C of a decision tree can be computed by the recursive formulation:

$$\mathrm{C}(\mathcal{D}, u) = \begin{cases} \sum_{(x,k) \in \mathcal{D}} \mathbb{1}(k \neq l(u)) & \text{if } u \in L \\ \mathrm{C}(\mathcal{D}_{\overline{b(u)}}, u_L) + \mathrm{C}(\mathcal{D}_{b(u)}, u_R) & \text{if } u \in B \end{cases} \tag{1}$$

The task is to find an optimal decision tree $\tau$ that minimizes the objective value over the training data given a maximum tree depth $d$. We now explain how this can be optimized with DP, and in the next section, we generalize this to any separable decision-tree optimization task.

**Dynamic programming**    DP simplifies a complex problem by reducing it to smaller repeated subproblems for which solutions are cached. For example, minimizing misclassification score with a

binary class can be solved with DP as follows (adapted from [20]):

$$T(\mathcal{D}, d) = \begin{cases} \min\{|\mathcal{D}^+|, |\mathcal{D}^-|\} & d = 0 \\ \min_{f \in \mathcal{F}} \left\{ T(\mathcal{D}_f, d-1) \; + \; T(\mathcal{D}_{\bar{f}}, d-1) \right\} & d > 0 \end{cases} \tag{2}$$

This equation computes the minimum possible misclassification score for a dataset $\mathcal{D}$ and a given maximum tree depth $d$. Recursively, as long as $d > 0$, a feature $f$ is selected for branching such that the sum of the misclassification score of the left and right subtree is minimal. In the leaf node (when $d = 0$), the label of the majority class is selected: either the positive ($\mathcal{D}^+$) or negative ($\mathcal{D}^-$) label. This approach is further optimized by using caching and bounds.

As is common with DP notation, Eq. (2) returns the solution value (or cost) and not the solution (the tree). In the remainder of the text, we refer to solution values and solutions interchangeably, with one implying the other and vice versa.

While Eq. (2) yields a single optimal solution, Demirović and Stuckey [19] investigated the use of DP for nonlinear bi-objective optimization by searching for a Pareto front of optimal solutions: i.e., the set of solutions that are not Pareto dominated ($\succ$) by any other solution:

$$\text{nondom}(\Theta) = \{v \in \Theta \mid \neg \exists \, v' \in \Theta \, (v' \succ v)\} \tag{3}$$

Consequently, every subtree search also no longer returns just one solution but a Pareto front. For this, they introduce a merge function that combines every solution from one subtree with every solution from the other subtree, resulting in a new Pareto front.

## 4 Framework for separable objectives

In this section, we generalize previous DP methods for optimal decision trees to a new general framework that can solve any separable optimization task. First, we define the problem. Second, we formalize what is meant by separability in the context of learning decision trees and then we prove necessary and sufficient conditions for optimization tasks to be separable. Third, we present the generalized framework. Finally, we show separable optimization task formulations for four example domains.

### 4.1 Problem definition and notation

We now generalize the previously introduced problem of minimizing misclassification score to any decision tree optimization task by using common DP notation. Given a state space $\mathcal{S}$ and a solution space $\mathcal{V}$, we define:

**Definition 4.1** (Optimization task). An *optimization task* is described by six components:

1. a cost function $g : \; \mathcal{S} \times (\mathcal{F} \cup \mathcal{K}) \to \mathcal{V}$ that returns the cost of action $a \in \mathcal{F} \cup \mathcal{K}$ in state $s \in \mathcal{S}$, where the action is either assigning label $\hat{k} \in \mathcal{K}$ or branching on feature $f \in \mathcal{F}$;
2. a transition function $t : \; \mathcal{S} \times \mathcal{F} \times \{0, 1\} \to \mathcal{S}$ that provides the next state after branching left or right on feature $f$, denoted by $f$ or $\bar{f} \in \mathcal{F} \times \{0, 1\}$;
3. a comparison operator $\succ \; : \; \mathcal{V} \times \mathcal{V} \to \{0, 1\}$ that determines Pareto dominance;
4. a combining operator $\oplus \; : \; \mathcal{V} \times \mathcal{V} \to \mathcal{V}$ that combines solution values to one value;
5. a constraint $c : \; \mathcal{V} \times \mathcal{S} \to \{0, 1\}$ that determines feasibility of a given solution $v$ and state $s$;
6. and an initial state $s_0 \in \mathcal{S}$.

For all optimization tasks in this paper, the state $s$ can be described by the dataset $\mathcal{D}$ and the branching decisions $F$ in parent nodes. The transition function is given by $t(\langle \mathcal{D}, F \rangle, f) = \langle \mathcal{D}_f, F \cup \{f\} \rangle$, and the initial state $s_0$ is $\langle \mathcal{D}, \emptyset \rangle$. But our framework extends beyond this as well.

When minimizing misclassifications, the cost function is $g(\langle \mathcal{D}, F \rangle, \hat{k}) = |\{(x, k) \in \mathcal{D} \mid k \neq \hat{k}\}|$; the comparison operator $\succ$ is $<$; the combining operator $\oplus$ is addition and the constraint $c$ returns all solutions as feasible. Similarly, more complex objectives, such as F1-score and group fairness, which do not fit the conditions of DL8 [66], can be defined by using these building blocks. We show four examples in Section 4.4.

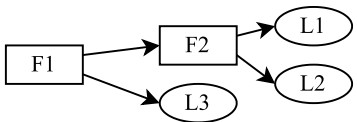

Figure 1: An example tree of depth two with five decision variables: two branching decisions F1 and F2, and three leaf node label assignments: L1, L2, and L3.

The final cost of a tree $\tau = (B, L, b, l)$ with root node $r$ is now given by calling $C(s_0, r)$ on the following recursive function, which generalizes Eq. (1):

$$C(s, u) = \begin{cases} g(s, l(u)) & \text{if } u \in L \\ C(t(s, \overline{b(u)}), u_L) \ \oplus \ C(t(s, b(u)), u_R) \ \oplus \ g(s, b(u)) & \text{if } u \in B \end{cases} \tag{4}$$

In leaf nodes, the cost of a tree is given by cost function $g$. In branching nodes, the combining operator $\oplus$ combines the cost of the left and right subtree and the branching costs.

Given an optimization task $o = \langle g, t, \succ, \oplus, c, s_0 \rangle$ and a maximum tree depth $d$, the aim is to find a tree (or a Pareto front of trees) that satisfies the constraint $c$ and optimizes (according to the comparison operator $\succ$) the cost C.

With this definition of an optimization task, we can generalize several of the concepts introduced in the preliminaries. Let $\Theta$ describe a set of possible solutions and let

$$\text{feas}(\Theta, s) = \{v \in \Theta \mid c(v, s) = 1\} \tag{5}$$

be the subset of all feasible solutions in $\Theta$ for state $s$. The optimal set of solutions is given by

$$\text{opt}(\Theta, s) = \text{nondom}(\text{feas}(\Theta, s)), \tag{6}$$

the Pareto front of feasible solutions in $\Theta$, with $\text{nondom}$ depending on the comparison operator $\succ$. Furthermore, we generalize the definition for $\text{merge}$ from [19]:

$$\text{merge}(\Theta_1, \Theta_2, s, f) = \{v_1 \oplus v_2 \oplus g(s, f) \mid v_1 \in \Theta_1, v_2 \in \Theta_2\} \tag{7}$$

This returns all combinations of solutions from sets $\Theta_1$ and $\Theta_2$ by using the combining operator $\oplus$.

## 4.2 Separability

An optimization task is separable if optimal solutions to subtrees can be computed independently of other subtrees. For example, in the tree in Fig. 1, the optimal label L3 should be independent of the branching decision F2 and labels L1 and L2. Label L2 should be independent of L1 and L3.

**Definition 4.2** (Separable). An optimization task is *separable* if and only if the optimal solution to any subtree can be determined independently of any of the decision variables that are not part of that subtree or the parent nodes' branching decisions.

The general idea to prove that an optimization task satisfies Def. 4.2 is to use complete induction over the maximum tree depth $d$. For the base step ($d = 0$) it is sufficient to require the cost and transition functions to be *Markovian*: their output should depend only on the current state and the chosen decision. For the induction step ($d > 0$), we need to show that the optimization task satisfies the *principle of optimality* [9]: that optimal solutions for trees of depth $d$ can be constructed from only the optimal solutions to its subtrees and the new decision (see Def. A.1). In summary:

**Proposition 4.3.** *An optimization task $o = \langle g, t, \succ, \oplus, c, s_0 \rangle$ is* separable *if and only if its cost function $g$ and transition function $t$ are* Markovian *and the task $o$ satisfies the* principle of optimality.

A full proof for this and all subsequent propositions and theorems can be found in Appendix A.

To satisfy the *principle of optimality*, the combining operator $\oplus$ must *preserve order* over $\succ$, and the constraint $c$ must be *anti-monotonic*. These two notions are explained next.

We introduce the new notion *order preservation* that guarantees that any combination of optimal solutions using the combining operator $\oplus$ always dominates a combination with at least one suboptimal solution. Many objectives, such as costs, are *additive* with $<$ as the $\succ$ operator. Addition preserves order over $<$. However, *additivity* is not a necessary condition. We present *order preservation* instead as a necessary condition (see Appendix A.2).

**Definition 4.4** (Order preservation). A combining operator $\oplus$ *preserves order* over a given comparator $\succ$, if for any given state $s$, feature $f$ and solution sets $\Theta_1$ and $\Theta_2$, with $s_1 = t(s, f)$, $s_2 = t(s, \bar{f})$, $v_1 \in \mathrm{opt}\,(\Theta_1, s_1)$, $v_1' \in \Theta_1$ and $v_2 \in \mathrm{opt}\,(\Theta_2, s_2)$, with $v_1 \succ v_1'$, then $v_1 \oplus v_2 \succ v_1' \oplus v_2$ (and $v_2 \oplus v_1 \succ v_2 \oplus v_1'$, if $\oplus$ is not commutative).

To guarantee that applying feas as defined in Eq. (5) does not prune any partial solution that is part of the final optimal solution, constraint $c$ must be *anti-monotonic* [66]: if a constraint is violated in a tree, it is also violated in any tree of which this tree is a subtree. However, in order to speak of a *necessary* condition, we redefine anti-monotonicity to require that any solution which is constructed from at least one subsolution that is infeasible, cannot be an optimal solution.

**Definition 4.5** (Anti-monotonic). A constraint $c$ is *anti-monotonic* if for any state $s$, feature $f$ and solution sets $\Theta_1$ and $\Theta_2$, with $s_1 = t(s, f)$, $s_2 = t(s, \bar{f})$, with $v_1 \in \Theta_1$ and $v_2 \in \Theta_2$, if $\neg c(v_1, s_1)$ or if $\neg c(v_2, s_2)$, then $v_1 \oplus v_2 \notin \mathrm{opt}\,(\Theta, s)$.

Minimum support (minimum leaf node size) is an example of an anti-monotonic constraint [66].

With these conditions and definitions in place, we have the necessary and sufficient conditions for separability and can present the main theoretical result of this paper:

**Theorem 4.6.** *An optimization task $o = \langle g, t, \succ, \oplus, c, s_0 \rangle$ is* separable *if and only if its cost function $g$ and transition function $t$ are* Markovian*, its combining operator $\oplus$ preserves order* over its comparison operator $\succ$ and the constraint $c$ is anti-monotonic.

### 4.3 Dynamic programming formulation

We now present the general DP framework *STreeD* (Separable Trees with Dynamic programming, pronounced as *street*):

$$T(s, d) = \begin{cases} \mathrm{opt}\left(\bigcup_{\hat{k} \in \mathcal{K}} \{\, g(s, \hat{k})\, \}, s\right) & d = 0 \\ \mathrm{opt}\left(\bigcup_{f \in \mathcal{F}} \mathrm{merge}\left(T(t(s, f), d - 1), T(t(s, \bar{f}), d - 1), s, f\right), s\right) & d > 0 \end{cases} \quad (8)$$

Pseudo-code for STreeD and additional algorithmic techniques for speeding up computation, such as a special depth-two solver, caching, and upper and lower bounds, as well as techniques for sparse trees and hypertuning, can be found in Appendix B. The appendix also provides the conditions for the use of these techniques. Furthermore, in Appendix A we prove the following Theorem:

**Theorem 4.7.** *STreeD, as defined in Eq.* (8)*, finds the Pareto front for any optimization task that is* separable *according to Def. 4.2*.

### 4.4 Examples of separable optimization tasks

To illustrate the flexibility of STreeD, we present four diverse example applications for which separable optimization tasks can be formulated. The first two are also covered by the *additivity* condition from [66]. The last two do not fit this condition but are covered by our framework.

**Cost-sensitive classification** Cost-sensitive classification considers costs related to obtaining (measuring) the value of features in addition to misclassification costs [55]. Typical applications are in the medical domain when considering expensive diagnostic tasks and asymmetrical misclassification costs. See Appendix C for a more comprehensive introduction and related work.

Let $\mathcal{D}$ be the instances in a leaf node and let $M_{k, \hat{k}}$ be the misclassification costs when an instance with true label $k$ is assigned label $\hat{k}$, and let $m(F, f)$ be the costs of obtaining the value of feature $f$ (this value also depends on the set of previously measured features $F$, because we consider that some tests can be performed in a group and share costs). This results in the following cost function:

$$g(\langle \mathcal{D}, F \rangle, \hat{k}) = \sum_{(x, k) \in \mathcal{D}} M_{k, \hat{k}} \qquad\qquad g(\langle \mathcal{D}, F \rangle, f) = |\mathcal{D}| \cdot m(F, f) \qquad (9)$$

This optimization task is separable because the function $g$ is Markovian, the combination operator $\oplus$ is addition, and the comparison operator is $<$.

**Prescriptive policy generation** Decision trees can also be used to prescribe policies based on historical data, to maximize the expected reward (revenue) of the policy [12, 45]. An example is medical treatment assignment based on historical treatment response. This is done by reasoning over counterfactuals (the expected outcome if an action other than the historical action was taken).

In Appendix D we provide related work and show three different methods for calculating the expected reward from the literature: regress and compare, inverse propensity weighting, and the doubly robust method. The cost function for each of these is Markovian and the combining operator is addition. Therefore, prescriptive policy generation can be formulated as a separable optimization task. It is also possible to add constraints to the policy, for example, when maximizing revenue under a capacity constraint. We show in Appendix A.4 that capacity constraints are also separable.

**Nonlinear classification metrics** Nonlinear objectives such as F1-score and Matthews correlation coefficient are typically used for unbalanced datasets. Demirović and Stuckey [19] provide a globally optimal DP formulation for optimizing such nonlinear objectives for binary classification. STreeD can also optimize for these metrics. For F1-score, for example, one can formulate a combined optimization task that measures the false positive rate for each class $k \in \{0, 1\}$ as a separate optimization task:

$$g_k(\mathcal{D}, \hat{k}) = \begin{cases} |\{(x, k') \in \mathcal{D} \mid k' \neq \hat{k}\}| & \hat{k} = k \\ 0 & \text{otherwise} \end{cases} \tag{10}$$

Both of these tasks are separable, and in Appendix A.5 we show that multiple separable tasks can be combined into a new separable optimization task that results in the Pareto front for misclassifications for each class. This Pareto front can then be used to find the decision tree with, e.g., the best F1-score. See Appendix E for more related work and other details.

**Group fairness** Optimizing for accuracy can result in unfair results for different groups and therefore several approaches have been recommended to prevent discrimination [60], one of which is demographic parity [28], which states that the expected outcome for two classes should be the same. Previous MIP formulations for group fairness have been formulated [1, 44] and recently also a DP formulation, even though at first hand a group fairness constraint does not seem separable [53].

Let $a$ denote a discrimination-sensitive binary feature and $y$ a binary outcome, with $y = 1$ the preferred outcome and $\hat{y}$ the predicted outcome; then demographic parity can be described as satisfying $P(\hat{y} = 1 \mid a = 1) = P(\hat{y} = 1 \mid a = 0)$. This requirement is typically satisfied by limiting the difference to some percentage $\delta$. Let $N(a)$ be the number of people in group $a$, and $N(\bar{a})$ the people not in group $a$. The demographic parity requirement can now be formulated as two separable threshold constraints:

$$g_a(\mathcal{D}, \hat{k}) = \begin{cases} \frac{|\mathcal{D}_a|}{N(a)} & \hat{k} = 1 \\ \frac{|\mathcal{D}_{\bar{a}}|}{N(\bar{a})} & \hat{k} = 0 \end{cases} \qquad g_{\bar{a}}(\mathcal{D}, \hat{k}) = \begin{cases} \frac{|\mathcal{D}_a|}{N(a)} & \hat{k} = 0 \\ \frac{|\mathcal{D}_{\bar{a}}|}{N(\bar{a})} & \hat{k} = 1 \end{cases} \tag{11}$$

These cost functions with the two threshold constraints that limit both to $1 + \delta$, and an accuracy objective can be combined into one separable optimization task as shown in Appendix A.4 and A.5, and therefore it can be solved to optimality with STreeD. We show the derivation of these constraints, a similar derivation for equality of opportunity, and more related work in Appendix F. Note that we require demographic parity on the whole tree and not on every subtree. In fact, requiring demographic parity on every subtree is an example of a constraint that does not satisfy anti-monotonicity.

### 4.5 Comparison to previous theory and methods

**Dynamic programming theory** In contrast to what is common in general DP theory, our theory does not assume solution values to be real valued [48] or totally ordered [29]. For example, Karp and Held [48] formulate the *monotonicity* requirement of a DP as follows: for a given state $s$, action $a$, cost $c$ and data $p$, let $h(c, s, a, p)$ be the cost of reaching state $t(s, a)$ through an input sequence that reaches state $s$ with cost $c$. Monotonicity holds iff $c_1 \leq c_2$ implies $h(c_1, s, a, p) \leq h(c_2, s, a, p)$. In words: if a partial input sequence $A_1$ with cost $c_1$ dominates another input sequence $A_2$ with $c_2$, then any other input sequence that starts with $A_1$ dominates any other input sequence that starts with $A_2$, thus satisfying the principle of optimality. This notion is similar to our *order preservation*, but we do not assume that the costs are real valued or that the costs are totally ordered.

Similarly, Li and Haimes [51] show how multiple optimization tasks can be combined into one, as we also show in Appendix A.5. They construct a separable DP formulation for optimization tasks that can be deconstructed into a series of optimization tasks, each of which is separable and monotonic. Their method also returns the set of nondominated solutions. However, their theory assumes that the solution value for each sub-objective is real, completely ordered, and additive. This limits their theory to element-wise additive optimization tasks. Our theory does not share these limitations.

**Dynamic programming approaches**   In comparison to the DP methods DL8 [66] and GOSDT [52], STreeD covers a wider variety of objectives and constraints. Both STreeD and DL8 require constraints to be *anti-monotonic*. GOSDT does not support constraints. DL8 and GOSDT require *additivity*, whereas STreeD requires the less restrictive condition *order preservation* (see Def. 4.4). Moreover, STreeD can cover a range of new objectives by also allowing for objectives and constraints with a partially defined comparison operator, and by allowing to combine several such objectives and constraints. Examples of problems that could not be solved to global optimality by DL8 and GOSDT, but can be solved with our framework, are group fairness constraints, nonlinear metrics, and revenue maximization under a capacity constraint. Moreover, compared to DL8 and GOSDT, STreeD implements several algorithmic techniques for increasing scalability, such as a specialized solver for trees up to depth two (see Appendix B).

**Other approaches**   MIP can be used to model many optimization tasks, including non-separable optimization tasks. For example, optimizing decision tree policies for Markov Decision Processes can be optimized with MIP [89], but it is not clear how to do this with DP. However, MIP cannot deal with separable, but non-linear objectives, such as F1-score. Moreover, as our experiments also show, DP outperforms the scalability of MIP for the considered optimization tasks by several orders of magnitude.

CP so far has only been applied to maximizing accuracy [84] and (Max)SAT only to maximizing accuracy [34, 76] or finding the smallest perfect tree [42, 64].

Dunn [26] proposes a framework that can optimize many objectives for decision trees (see also [11]). Because of MIP's scalability issues, they propose a local search method based on coordinate descent. However, this method does not guarantee a globally optimal solution. Moreover, their method is not able to find a Pareto front that optimizes multiple objectives at the same time.

## 5   Experiments

To show the flexibility of STreeD, we test it on the four example domains introduced in Section 4.4. Additionally, we compare STreeD with two state-of-the-art dynamic programming approaches for maximizing classification accuracy. STreeD is implemented in C++ and is available as a Python package.[1] All experiments are run on a 2.6 GHz Intel i7 CPU with 8GB RAM using only one thread. MIP models are solved using Gurobi 9.0 with default parameters [33].

In our analysis, we focus on scalability performance in comparison to the state-of-the-art optimal methods (if available). The following shows a summary of the results. The setup of all experiments, more extensive results, and related work of the application domains is provided in the appendices C-G.

### 5.1   Cost-sensitive classification

For cost-sensitive classification, we compare STreeD with TMDP [58] both in terms of average expected costs and scalability. A direct fair comparison with TMDP is difficult, because TMDP provides no guarantee of optimality, allows for multi-way splits, and is run without a depth limit. We test STreeD and TMDP on three setups for 15 datasets from the literature and report normalized costs. We tune STreeD with a depth limit of $d = 4$. In Appendix C we show extended results and also compare with several heuristics from the literature. To our knowledge, no MIP methods for cost-sensitive classification including group discounts (i.e., when certain features are tested together, the feature cost decreases) exist, and therefore are not considered here.

For 21 out of 45 scenarios STreeD generates trees with significantly lower average out-of-sample costs than TMDP ($p$-value $< 5\%$). TMDP has lower costs in 1 out of 45 scenarios. In all other scenarios,

---

[1] https://github.com/AlgTUDelft/pystreed

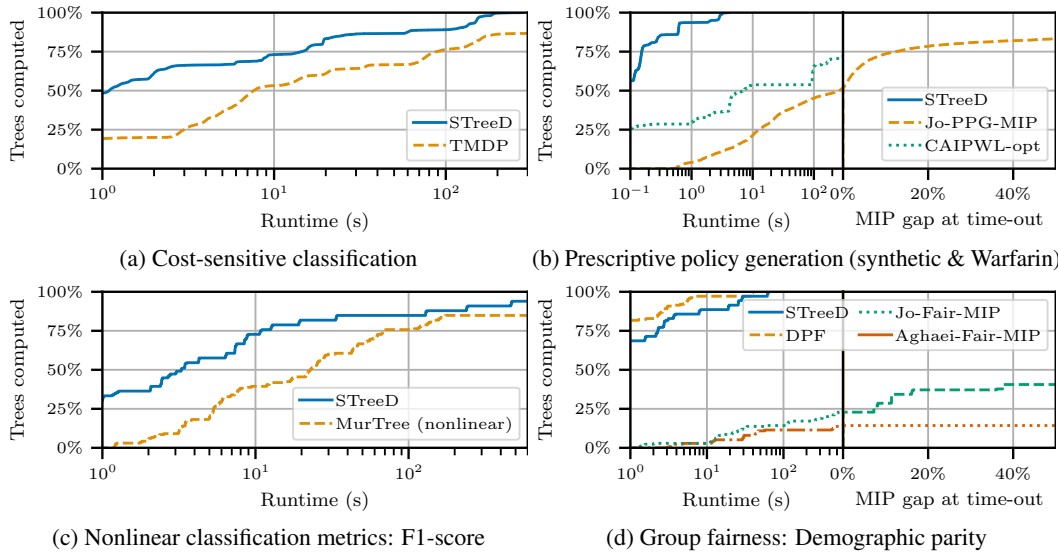

(a) Cost-sensitive classification

(b) Prescriptive policy generation (synthetic & Warfarin)

(c) Nonlinear classification metrics: F1-score

(d) Group fairness: Demographic parity

Figure 2: The percentage of instances for which the algorithm returns the (optimal) tree within the given runtime (higher is better). When MIP methods hit the timeout, the plot shows which percentage of problems are solved within the reported MIP gap. Instances for which all methods return a tree within one second are left out. Note the logarithmic x-axis.

there is no significant difference. Both methods return trees that perform better and are smaller than those constructed by heuristics. Fig. 2a shows that STreeD is significantly faster than TMDP. Thus, it can be concluded that STreeD on average has slightly better out-of-sample performance than TMDP, returns trees of similar size, and is significantly faster than TMDP.

## 5.2 Prescriptive policy generation

For prescriptive policy generation, we compare our method with the MIP model Jo-PPG-MIP [43] and the recursive tree search algorithm CAIPWL-opt [92]. Jo et al. [43] show both analytically and through experiments the benefit of their method over [12] and [45]. Their method constructs optimal binary trees based on three different teacher models: regress and compare, inverse propensity weighting, and a doubly robust method. CAIPWL-opt constructs optimal binary trees based on the doubly robust method. Our method implements the same teacher models; thus, all methods find the same solution. Therefore, we compare here only the runtime performance.

As described in detail in Appendix D, we test on the Warfarin dataset [39] and generate 225 training scenarios with 3367 instances and 29 binary features and run each algorithm with a given maximum depth of $d \in [1, 5]$. We also generate 275 synthetic datasets with 10, 100, and 200 binary features, each with 500 training samples, and run each algorithm on every scenario with a given maximum depth of $d = 1, 2$ and $3$. When Jo-PPG-MIP hits the timeout of 300 seconds, we report the MIP gap.

Fig. 2b shows that STreeD can find optimal solutions orders of magnitude faster than both Jo-PPG-MIP and CAIPWL-opt while obtaining the same performance in maximizing the expected outcome.

## 5.3 Nonlinear classification metrics

For assessing STreeD's performance on optimizing nonlinear metrics such as F1-score, we first compare it with the existing nonlinear MurTree method [19]. We tasked both methods to find optimal trees of depth 3-4 according to an F1-score metric for 25 binary classification datasets from the literature. Fig. 2c shows that STreeD, while being more general, is on average seven times faster than MurTree (computed with geometric mean). This performance gain is in part due to a more efficient computation of the lower and upper bounds, as explained in Appendix B.

This shows that STreeD can be applied to nonlinear objectives and even improves on the performance of a state-of-the-art specialized DP method for this task.

### 5.4   Group fairness

For group fairness, we follow the experiment setup from [53] as explained in Appendix F. The task is to search for optimal trees for 12 datasets and for trees of depth $d \in [1, 3]$ under a demographic parity constraint with at most $1\%$ discrimination on the training data. We compare to the DP-based method DPF [53] and the MIP methods Aghaei-Fair-MIP [1] and Jo-Fair-MIP [44]. When the MIP methods hit the timeout of 600 seconds, we report the MIP gap.

Fig. 2d shows the runtime results. As expected, DPF performs best since it is a DP method specifically designed for this task. STreeD is more general and yet remains close to DPF, while being several orders of magnitude faster than both MIP methods. In Appendix F we further show how STreeD is the first optimal DP method for optimizing an equality-of-opportunity constraint.

It can be concluded that STreeD allows for easy modeling of complex constraints such as demographic parity and equality of opportunity while outperforming MIP methods by a large margin.

### 5.5   Classification accuracy

Finally, we compare STreeD with two state-of-the-art dynamic programming approaches for optimal decision trees that maximize classification accuracy: DL8.5 [3, 4] and MurTree [20]. All three methods find optimal decision trees for maximum depth four and five for 25 binary classification datasets from the literature. The results in Appendix G show that on average STreeD is more than six times faster than DL8.5 and is close to MurTree, with MurTree being 1.25 faster (computed with the geometric mean). Since DL8.5 outperforms DL8 [3] and MurTree outperforms GOSDT by a large margin [20], we can conclude that STreeD not only allows for a broader scope of optimization tasks than DL8 and GOSDT, but is also more scalable.

## 6   Conclusion

We present STreeD: a general framework for finding optimal decision trees using dynamic programming (DP). STreeD can optimize a broad range of objectives and constraints, including nonlinear objectives, multiple objectives, and non-additive objectives, provided that these objectives are separable. We apply DP theory to provide necessary and sufficient requirements for such separability in decision tree optimization tasks and introduce the new notion order preservation that guarantees the principle of optimality [9]. We show that these results are more general than the state-of-the-art general DP solvers for decision trees [52, 66].

We illustrate the generalizability of STreeD in five application domains, two of which are not covered by the previous general DP solvers. For each, we present a separable optimization formulation and compare STreeD's scalability to the domain-specific state-of-the-art. The results show that STreeD sustains or improves the scalability performance of DP methods, while being flexible, and outperforms mixed-integer programming methods by a large margin on these example domains.

STreeD allows for a much broader set of applications than presented here. Therefore, future work should further investigate STreeD's performance on, for example, regression tasks and other non-additive objectives. STreeD could also be used to further develop optimization of trees that lack explanation redundancy [40]. Finally, STreeD could further be improved by adding multi-way branching and allowing for continuous features.

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

# A  Proofs

The following sections provide proofs for the propositions and theorems from the main text. We first prove Prop. 4.3 that claims that an optimization task is separable if it has a Markovian cost and transition function and if it satisfies the principle of optimality. We then prove Theorem 4.6 about the necessary and sufficient conditions for an optimization task to be separable. Next, we prove Theorem 4.7 about STreeD being able to solve separable optimization tasks to optimality. We extend the theory of the main text by introducing anti-monotonic threshold constraints. Finally, we show how multiple separable optimization tasks can be combined into a new separable optimization task.

## A.1  Proof of Prop. 4.3 about Separability

Def. 4.2 defines separability by stating that for a given optimization task the optimal solution to any subtree in a tree search should be independent of any of the decision variables other than the decision variables of the subtree search and the parent node branching decisions. Prop. 4.3 states that this is the same as saying that the optimization task has a Markovian cost and transition function and satisfies the principle of optimality [9], i.e., optimal solutions can only be constructed from optimal solutions to subproblems:

**Definition A.1** (Principle of Optimality). Let $o = \langle g, t, \succ, \oplus, c, s_0 \rangle$ be an optimization task, let $s$ be a state, let $f$ be a branching decision, and let $\Theta_1$ and $\Theta_2$ be all the solutions to subproblems with state $t(s, f)$ and $t(s, \bar{f})$, and let $\Theta = \mathrm{merge}(\Theta_1, \Theta_2, s, f)$. Then optimization task $o$ satisfies the *principle of optimality* if and only if for every state $s$, branching decision $f$ and solution sets $\Theta_1$ and $\Theta_2$, the following holds:

$$\mathrm{opt}\,(\Theta, s) = \mathrm{opt}\,\big(\mathrm{merge}(\mathrm{opt}\,(\Theta_1, t(s, f))\,, \mathrm{opt}\,\big(\Theta_2, t(s, \bar{f})\big)\,, s, f)\big) \tag{12}$$

With this definition in place, we can prove Prop. 4.3.

*Proof.* Prop. 4.3 requires two conditions: a Markovian cost and transition function, and the principle of optimality. To prove Prop. 4.3, we need to show that both conditions are necessary and sufficient. Therefore, we first show that if both conditions hold, the optimization task must be separable. We then show that if one of the two conditions of Prop. 4.3 fails, the optimization task is not separable.

*Sufficient:*  To show that the two conditions of Prop. 4.3 are sufficient we use total induction over the tree depth $d$.

The base step ($d = 0$), considers the search for optimal leaf nodes. If a Markovian cost function $g(s, \hat{k})$ exists, the optimal label assignment can be determined by $\mathrm{opt}\,\big(\cup_{\hat{k} \in \mathcal{K}} g(s, \hat{k}), s\big)$. This does not depend on any decision variable other than the state $s$ and the leaf node label assignment $\hat{k}$ and therefore Def. 4.2 is satisfied for all tree searches up to depth zero.

The induction step (for $d > 0$) assumes Def. 4.2 is satisfied for all tree searches up to depth $d - 1$. For these tree searches, optimal solutions can now be constructed by iterating over all possible branching decisions for the root node, using Eq. (12) to find optimal solutions when the root node branching decision is fixed, and using $\mathrm{opt}$ to select the optimal solution from the union of all these solution sets. Since neither the subtree search, nor Eq. (12), nor the transition function, nor the branching cost function, nor the union operator, nor the $\mathrm{opt}$ function depend on any decision variable other than the decision variables for this tree search and the parent node decisions, the optimal solution for this search can also be constructed independently of those variables.

*Necessary conditions:*  If an optimization task is separable, a Markovian cost function and transition function necessarily exist. Since a node's state is the product of applying the transition function on the previous state, this state can be considered as a function of the initial state $s_0$, the parent nodes' branching decisions $F$, and the immediate action: the branching decision or the label assignment. Now assume that no Markovian cost and transition function exist. This means that the cost of an action cannot be expressed in terms of only the state, the immediate action, and the parent's branching decisions. This means that Def. 4.2 is not satisfied: a contradiction. Therefore a Markovian cost and transition function necessarily exists if an optimization task is separable.

For tree searches with a depth larger than zero, optimal solutions should be able to be constructed from optimal solutions to subtree searches, as defined in Eq. (12). Consider such a subtree search,

which after deciding on the branching decision in its root node, creates two subproblems for the resulting children nodes. The optimal solution should be constructible from the optimal solution to these subproblems according to Eq. 12. If the optimal solution to this tree search cannot be constructed from the optimal solution to its subtree searches, then the optimal solutions to the subtree searches are not optimal: a contradiction. Therefore Eq. (12) is a necessary condition. $\square$

## A.2 Proof of Theorem 4.6 about Conditions for Separability

Theorem 4.6 states that an optimization task is separable if its cost and transition functions are Markovian, its combining operator preserves order over the comparison operator and its constraint is anti-monotonic.

To prove that these conditions are both sufficient and necessary, we need to show that the conditions mentioned in Theorem 4.6 are equivalent to those mentioned in Prop. 4.3. We will first show that these conditions are sufficient, and then that they are also necessary.

*Proof.* Let $o = \langle g, t, \succ, \oplus, c, s_0 \rangle$ be an optimization task, for which it holds that the cost function $g$ and transition function $t$ is Markovian, the combining operator $\oplus$ preserves order over $\succ$ and the constraint $c$ is anti-monotonic.

We will call an optimization task that satisfies Eq. (12) *splittable*. We begin by proving that $o$ is splittable. Consider any state $s$, any branching feature $f$, such that $s_1 = t(s, f)$ and $s_2 = t(s, \bar{f})$, and let $\Theta, \Theta_1$ and $\Theta_2$ be sets of solutions such that $\Theta = \mathrm{merge}(\Theta_1, \Theta_2, s, f)$, and let the optimal solutions from these sets be given by $\theta = \mathrm{opt}(\Theta, s)$, $\theta_1 = \mathrm{opt}(\Theta_1, s_1)$ and $\theta_2 = \mathrm{opt}(\Theta_2, s_2)$. Then, according to Eq. (12) an optimization task is splittable if and only if:

$$\theta = \mathrm{opt}(\mathrm{merge}(\theta_1, \theta_2, s, f), s) \tag{13}$$

For this to hold the solution set $\theta$ must be both a subset and a superset of the right-hand side of Eq. (13). We consider both next. However, for brevity of notation, we leave out the states $s, s_1, s_2$ and the branching feature $f$.

1. $\theta$ is a subset:
$$\theta \subseteq \mathrm{opt}(\mathrm{merge}(\theta_1, \theta_2)) \tag{14}$$

   By definition $\mathrm{opt}(\Theta) \subseteq \Theta$, and therefore Eq. (14) is always true if

   $$\theta \subseteq \mathrm{merge}(\theta_1, \theta_2) \tag{15}$$

   This can be shown by contradiction: Assume that Eq. (15) is false. This means there must exist a solution $v \in \theta$ for which $v \notin \mathrm{merge}(\theta_1, \theta_2)$. By definition $\theta = \mathrm{opt}(\Theta) = \mathrm{opt}(\mathrm{merge}(\Theta_1, \Theta_2))$. This means that this solution can be written as $v = v_1 \oplus v_2$, with $v_1 \in \Theta_1$ and $v_2 \in \Theta_2$. Because of the definition of merge, either $v_1 \notin \theta_1$ or $v_2 \notin \theta_2$. Assume w.l.o.g. that this is true for $v_1$. If $v_1 \notin \theta_1$, this is either because it is filtered out by feas or by nondom.

   (a) If $v_1$ is filtered out by feas, then $v_1 \notin \mathrm{feas}(\Theta_1)$. This means that $\neg c(v_1)$. Because $o$ is anti-monotonic, then by Def. 4.5 also $v_1 \oplus v_2 \notin \theta$, which contradicts the assumption made before.

   (b) If $v_1$ is filtered out by nondom, then $\exists v_1' \in \theta_1 : v_1' \succ v_1$. Now construct solution $v' = v_1' \oplus v_2$. Because $\oplus$ preserves order over $\succ$, it holds that $v' \succ v$. This means that $v \notin \theta$, contradicting our assumption.

   In either case, the assumption is contradicted, and therefore Eq. (14) must be true.

2. $\theta$ is a superset:
$$\theta \supseteq \mathrm{opt}(\mathrm{merge}(\theta_1, \theta_2)) \tag{16}$$

   Again, by contradiction, assume that Eq. (16) is false. This means there must exists a solution $v = v_1 \oplus v_2$, with $v \in \mathrm{opt}(\mathrm{merge}(\theta_1, \theta_2))$, such that $v \notin \theta$. This means that $v$ is either filtered out of $\theta$ by feas or by nondom.

(a) If $v$ is filtered out by feas, it cannot be a member of $\text{opt}(\text{merge}(\theta_1, \theta_2))$, since it also applies feas.

(b) If $v$ is dominated by another solution in $\theta$, then there must exist a solution $v' \in \theta : v' \succ v$, but $v' \notin \text{opt}(\text{merge}(\theta_1, \theta_2))$. However, because of Eq. (14), this cannot be the case.

Again, in either case, the assumption is contradicted, and therefore Eq. (16) is true.

Since both Eq. (14) and Eq. (16) are true, Eq. (13) must be true and thus $o$ is splittable.

Since both Prop. 4.3 and Theorem 4.6 share the conditions that the cost and transition function must be Markovian, and because $o$'s cost and transition function are Markovian and $o$ is splittable, optimization task $o$ is separable by Def. 4.2. Since this holds for any optimization task $o$ that satisfies the conditions of Theorem 4.6, these conditions are sufficient to prove separability.

The conditions are also necessary. The condition of a Markovian cost and transition function is shared among Prop. 4.3 and Theorem 4.6 and therefore obviously necessary. Therefore, what is left to show is that order preservation and anti-monotonic constraints are necessary conditions to show the principle of optimality. For brevity of notation, we again leave out the states $s, s_1, s_2$ and the branching feature $f$.

**Order preservation** Let $\Theta = \text{merge}(\Theta_1, \Theta_2)$, with $\theta = \text{opt}(\Theta), \theta_1 = \text{opt}(\Theta_1)$ and $\theta_2 = \text{opt}(\Theta_2)$. Now assume that the splitting property holds: This means that for all $v = v_1 \oplus v_2$, with $v \in \theta$, also $v_1 \in \theta_1$ and $v_2 \in \theta_2$ and the other way around. Now assume that order preservation is not necessary. This means $\exists v_1' \notin \theta_1$, with $v' = v_1' \oplus v_2$. Consider two cases: if $v' \in \theta$, then $v_1' \in \theta_1$: a contradiction. However, if $v' \notin \theta$, then also $v \notin \theta$: a contradiction.

**Anti-monotonicity** If the constraint $c$ is not anti-monotonic, then there exists a solution $v = v_1 \oplus v_2$ with $\neg c(v_1)$, but with $v \in \text{opt}(\Theta)$. If $\neg c(v_1)$, then $v_1 \notin \text{opt}(\Theta_1)$. Thus, the splitting property is not satisfied.

Since each condition separately is necessary and all conditions together are sufficient, Theorem 4.6 holds. $\qquad\square$

## A.3 Proof of Theorem 4.7 about the Optimality of STreeD

Theorem 4.7 states that any *separable* optimization task can be solved to optimality using Eq. (8) from the main text. Here, we provide a proof for this theorem.

*Proof.* Let $\Theta(s, d)$ denote the set of all possible solution values to a decision tree search problem with state $s$, and $d$ the maximum depth of the tree, and let $\theta(s, d)$ be the set of nondominated and feasible solutions for a given separable optimization task $o = \langle g, t, \succ, \oplus, c, s_0 \rangle$:

$$\theta(s, d) = \text{opt}(\Theta(s, d), s) \tag{17}$$

We need to prove that for every state $s$ and depth $d$, Eq. (8) returns the optimal solution:

$$\theta(s, d) = T(s, d) \tag{18}$$

We will prove this by induction over $d$.

*Base step:* If $d = 0$, because the cost function $g$ is Markovian, the solution set $\Theta(s, 0)$ can be described as:

$$\Theta(s, 0) = \bigcup_{\hat{k} \in \mathcal{K}} \left\{ g(s, \hat{k}) \right\} \tag{19}$$

Therefore the optimal solution set as defined by Eq. (17) is precisely what Eq. (8) returns for $d = 0$:

$$\theta(s, 0) = T(s, 0) \tag{20}$$

Therefore, since we made no further assumptions, the base step of this proof holds.

*Induction step:* For $d > 0$, we assume that Eq. (8) returns the optimal solution for $d - 1$:

$$\theta(s, d - 1) = T(s, d - 1) \tag{21}$$

We now need to show that Eq. (8) when $d > 0$ returns the optimal solution:

$$\theta(s, d) = \text{opt}(\Theta(s, d)) = \text{opt}\left(\bigcup_{f \in \mathcal{F}} \text{merge}\left(T(t(s, f), d - 1), T(t(s, \bar{f}), d - 1)\right)\right) \tag{22}$$

Define $\Theta_f(s, d)$ as denoting the set of all possible solution values with feature $f$ as the branching feature of the root:

$$\Theta_f(s, d) = \text{merge}\left(\Theta(t(s, f), d - 1), \Theta(t(s, \bar{f}), d - 1), s, f\right) \tag{23}$$

When $d > 0$, the set $\Theta(s, d)$ can be described as the combination of all sets :

$$\Theta(s, d) = \bigcup_{f \in \mathcal{F}} \Theta_f(s, d) \tag{24}$$

Any solution that would be filtered out in a subset would also be filtered out in the whole set, therefore:

$$\text{opt}(\cup_{\Theta'}\Theta') = \text{opt}(\cup_{\Theta'} \text{opt}(\Theta')) \tag{25}$$

We can now derive the following:

$$\begin{aligned}\theta(s, d) &= \text{opt}(\Theta(s, d))\\ &= \text{opt}\left(\cup_{f \in \mathcal{F}}\Theta_f(s, d)\right)\\ &= \text{opt}\left(\cup_{f \in \mathcal{F}} \text{opt}\left(\Theta_f(s, d)\right)\right)\end{aligned} \tag{26}$$

Because of Eq. (26), in order to prove Eq. (22), we only need to show the following:

$$\theta_f(s, d) = \text{opt}\left(\text{merge}\left(T(t(s, f), d - 1), T(t(s, \bar{f}), d - 1), s, f\right), s\right) \tag{27}$$

Since the induction step assumes $T$ computes optimal trees for $d - 1$, we may here substitute with Eq. (21):

$$\theta_f(s, d) = \text{opt}\left(\text{merge}\left(\theta(t(s, f), d - 1), \theta(t(s, \bar{f}), d - 1), s, f\right), s\right) \tag{28}$$

Because the optimization task is separable, it is splittable, and therefore Eq. (27) holds.

Therefore the induction step also holds. Since this proof is shown without any further assumptions, by mathematical induction Theorem 4.7 is true. $\qquad\square$

## A.4 Threshold constraints

Many constraints, such as capacity constraints, can be considered as an objective with a threshold value $\beta$:

$$c_\beta(v, s) = \mathbb{1}(v \succ \beta) \tag{29}$$

Eq. (29) means that a solution is feasible if its solution value is better ($\succ$) than the threshold. E.g., when minimizing, any solution with a value lower than the bound is feasible.

Constraints need to be anti-monotonic to be separable. Threshold constraints are anti-monotonic provided that the combining operator $\oplus$ is worsening and the comparison operator $\succ$ is transitive:

**Definition A.2.** (Worsening operator) A combining operator $\oplus$ is called *worsening* if and only if $v = v_1 \oplus v_2 \rightarrow v_1 \succeq v \wedge v_2 \succeq v$.

For example, when setting prices to maximize revenue under a maximum supply constraint, you can calculate the expected demand in each node, sum demand from several nodes by using addition, and then prune trees that exceed this maximum supply by using a threshold constraint.

**Proposition A.3.** *Given an optimization task $o = \langle g, t, \succ, \oplus, c_\beta, s_0 \rangle$, if the combining operator $\oplus$ satisfies the* worsening *property and if the comparison operator $\succ$ is* transitive*, then the threshold constraint $c_\beta$ is* anti-monotonic.

*Proof.* Given any $v_1 \in \Theta_1, v_2 \in \Theta_2$ with $\beta \succ v_1$ (or w.l.o.g., $\beta \succ v_2$), such that $v_1 \notin \theta_1$. Let $v = v_1 \oplus v_2$ with $\oplus$ worsening, then $\beta \succ v_1 \succeq v$ and therefore $v \notin \theta$, and therefore $c_\beta$ is anti-monotonic. $\qquad\square$

## A.5 Combining multiple optimization tasks

We now show that a combination of separable optimization tasks is also separable. This is important for multi-objective optimization or for optimizing with a constraint. For example, we can maximize revenue in the first task, while respecting a maximum supply constraint in the second task. Let $o_a = \langle g_a, t_a, \succ_a, \oplus_a, c_a, s_0^a \rangle$ and $o_b = \langle g_b, t_b, \succ_b, \oplus_b, c_b, s_0^b \rangle$ be two separable optimization tasks. Two (or more) of such tasks can now be combined with the following equations:

$$g(\langle s_a, s_b \rangle, \hat{k}) = \langle g_a(s_a, \hat{k}), g_b(s_b, \hat{k}) \rangle \tag{30}$$

$$g(\langle s_a, s_b \rangle, f) = \langle g_a(s_a, f), g_b(s_b, f) \rangle \tag{31}$$

$$t(\langle s_a, s_b \rangle, f) = \langle t_a(s_a, f), t_b(s_b, f) \rangle \tag{32}$$

$$\langle v_1^a, v_1^b \rangle \succ \langle v_2^a, v_2^b \rangle \text{ iff } \langle v_1^a, v_1^b \rangle \neq \langle v_2^a, v_2^b \rangle \wedge v_1^a \succeq v_2^a \wedge v_1^b \succeq v_2^b \tag{33}$$

$$\langle v_1^a, v_1^b \rangle \oplus \langle v_2^a, v_2^b \rangle = \langle v_1^a \oplus_a v_2^a, v_1^b \oplus_b v_2^b \rangle \tag{34}$$

$$c(\langle v_a, v_b \rangle, \langle s_a, s_b \rangle) = \mathbb{1}(c_a(v_a, s_a) \wedge c_b(v_b, s_b)) \tag{35}$$

$$s_0 = \langle s_0^a, s_0^b \rangle \tag{36}$$

Eqs. (30)-(31) define the cost function as returning a tuple of the values that $g_a$ and $g_b$ return. Eq. (32) returns the result of both transition functions. Eq. (33) gives a partially defined comparison operator which states that a solution only dominates another solution if the solution values for both optimization tasks dominate both values of another solution. Eq. (34) defines the combination operator by applying the combination operator of the sub-tasks to the corresponding values of the left and right-hand side. Eq. (35) defines a new constraint that is only feasible when the solution is feasible according to both sub-constraints. Finally, Eq. (36) sets the starting state as the combination of both starting states.

**Proposition A.4.** *Let $o_a = \langle g_a, t_a, \succ_a, \oplus_a, c_a, s_0^a \rangle$ and $o_b = \langle g_b, t_b, \succ_b, \oplus_b, c_b, s_0^b \rangle$ be two separable optimization tasks. Then the combined optimization task $\langle g, t, \succ, \oplus, c, s_0 \rangle$ defined by Eqs. (30)-(36) is also separable.*

To prove Prop. A.4 it is necessary to show that the optimization task resulting from applying Eqs. (30)-(36) satisfies all the properties of being a separable optimization task.

*Proof.* Given two separable optimization tasks $o_a = \langle g_a, t_a, \succ_a, \oplus_a, c_a, s_0^a \rangle$ and $o_b = \langle g_b, t_b, \succ_b, \oplus_b, c_b, s_0^b \rangle$, and an optimization task $o = \langle g, t, \succ, \oplus, c, s_0 \rangle$ which is the result of applying Eqs. (30)-(36) to $o_a$ and $o_b$.

*Markovian:* If $g_a, g_b, t_a$ and $t_b$ are Markovian, then $g$ and $t$ are also Markovian since their value can be fully expressed by calling the functions of the subtasks, which are Markovian, without considering any other decision variables.

*Order preservation:* If $\oplus_a$ preserves order over $\succ_a$ and $\oplus_b$ preserves order over $\succ_b$, then $\oplus$ also preserves order over $\succ$, because of the following.

Consider solutions $\langle v_1^a, v_1^b \rangle \in \text{opt}(\Theta_1, s_1)$, $\langle v_1^{a'}, v_1^{b'} \rangle \in \Theta_1$, so that $\langle v_1^a, v_1^b \rangle \succ \langle v_1^{a'}, v_1^{b'} \rangle$, consider $\langle v_2^a, v_2^b \rangle \in \text{opt}(\Theta_2, s_2)$ and assume that $\oplus$ is commutative. Then, because of Eq. (33), there are two options: either $v_1^a \succ_a v_1^{a'} \wedge v_1^b \succeq_b v_1^{b'}$, or $v_1^a \succeq_a v_1^{a'} \wedge v_1^b \succ_b v_1^{b'}$. W.l.o.g., choose the first. Since $\oplus_a$ preserves order over $\succ_a$ and $\oplus_b$ preserves order over $\succ_b$, we can now say that $v_1^a \oplus_a v_2^a \succ_a v_1^{a'} \oplus_a v_2^a$ and also $v_1^b \oplus_b v_2^b \succeq_b v_1^{b'} \oplus_b v_2^b$. Because of Eqs. (33)-(34), it follows:

$$\langle v_1^a, v_1^b \rangle \oplus \langle v_2^a, v_2^b \rangle \succ \langle v_1^{a'}, v_1^{b'} \rangle \oplus \langle v_2^a, v_2^b \rangle \tag{37}$$

Therefore $\oplus$ also preserves order over $\succ$.

If $\oplus$ is not commutative, the proof above should be repeated with the operands for every use of $\oplus$ switched around.

*Anti-monotonicity:* If $c_a$ and $c_b$ are both anti-monotonic, then for any given solution $v_1 = \langle v_1^a, v_1^b \rangle$ and $v_2$, and $v = \langle v^a, v^b \rangle = v_1 \oplus v_2$, and any given state $s$ and branching decision $f$, such that $s_1 = t(s, f)$ and $s_2 = t(s, \bar{f})$: if $\neg c_a(v_1^a, s_1)$ (or similarly for $c_b$ and $v_2$ and $s_2$), then also $\neg c_a(v^a, s)$, and therefore $\neg c(v, s)$, thus $c$ is also anti-monotonic.

*Conclusion* Since the combined optimization task $o$ satisfies all necessary properties, $o$ is also separable. $\square$

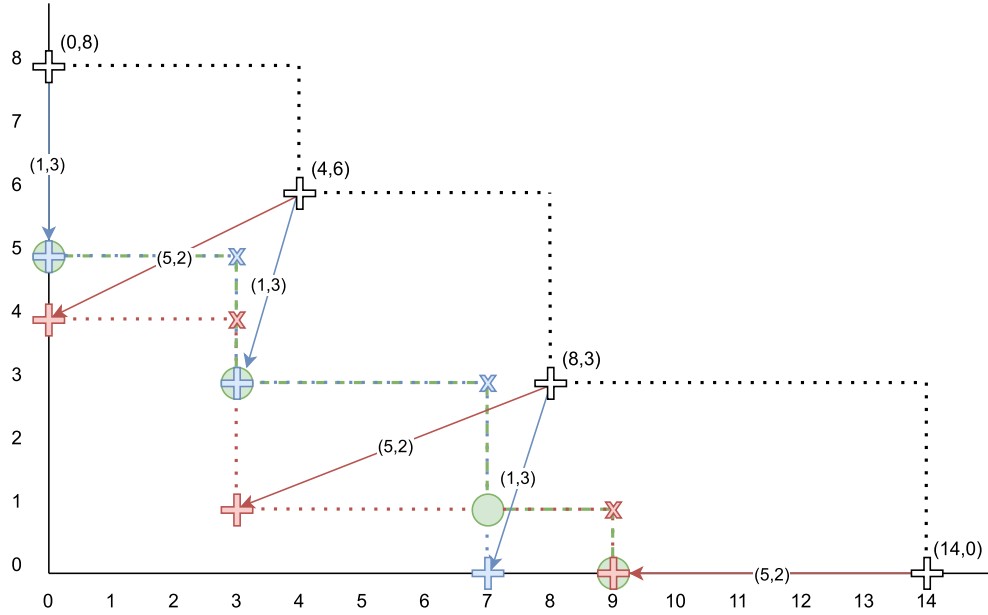

Figure 3: The result of subtracting the left solutions $\{(1,3),(5,2)\}$ from the upper bound $\{(0,8),(4,6),(8,3),(14,0)\}$. Subtracting the solution $(1,3)$ results in the blue Pareto front $\{(0,5),(3,3),(7,0)\}$. Subtracting the solution $(5,2)$ results in the red Pareto front $\{(0,4),(3,1),(9,0)\}$. Combining these two Pareto fronts results in the green Pareto front $\{(0,5),(3,3),(7,1),(9,0)\}$.

# B  Pseudo-code and Implementation

This section gives further details on the implementation of STreeD. First, we explain how upper and lower bounding works. Second, we expand on how to search for sparse trees and how to tune the number of nodes. Third, we provide conditions for caching. Fourth, pseudo-code for STreeD is given. Finally, a special solver for trees of depth two is discussed.

**Upper bounds**   A set of solutions ub is an upper bound to another set $\Theta$ if for all solution $v$ in $\Theta$ no solution in ub exists that is equal or dominates it:

$$\forall v \in \Theta : \neg \exists v' \in \text{ub} : v' \succeq v \tag{38}$$

An upper bound can be used to prune solutions that are dominated by it. The upper bound ub dominates a solution $v$ if there exists a solution $v' \in \text{ub}$ that is equal to or dominates $v$:

$$\text{ub} \leq v \Leftrightarrow \exists v' \in \text{ub} : v' \succeq v \tag{39}$$

This upper bound is used to prune poor solutions earlier in the search. To signify this we introduce the following new notation:

$$\text{opt}(\Theta, s, \text{ub}) = \text{opt}\left(\{v \in \Theta \mid v \leq \text{ub}\}, s\right) \tag{40}$$

**Subtracting**   In a branching node, when solutions $\Theta_1$ are known for one subtree, these can be used to update the upper bounds for the other subtree, provided that an appropriate *subtraction* operator $\ominus$ exists. For totally ordered optimization tasks, the upper bound for the other subtree can be simply computed by subtracting the solution from the current upper bound. However, for partially ordered objectives, the procedure is more complicated.

Consider the example shown in Fig. 3 where the goal is to minimize both dimensions, and the subtraction $\ominus$ is defined as element-wise subtraction (but values below zero are set to zero). The idea is to find first for each solution $v \in \Theta_1$ the upper bound $\text{ub}_v$ when $v$ is subtracted from the upper bound ub (the blue and red plusses in Fig. 3).

$$\text{ub}_v = \text{nondom}(\{u \ominus v \mid u \in \text{ub}\}) \tag{41}$$

The final new upper bound $\mathrm{ub}_2$ is the Pareto front that covers precisely the outer edge of all upper bounds $\mathrm{ub}_v$, such that if any solution is dominated by all $\mathrm{ub}_v$, it is dominated by $\mathrm{ub}_2$.

To compute this new upper bound, we need to define some more functions. Let $\prec$ decide if one solution is reverse Pareto dominated. Similarly, let $\overline{\mathrm{nondom}}$ be the set of reverse nondominated solutions. Let the operator $\triangle$ combine two solutions $v_1$ and $v_2$ such that $v_1 \triangle v_2 \preceq v_1$ and also $v_1 \triangle v_2 \preceq v_2$. Similarly, let the operator $\triangledown$ combine two solutions $v_1$ and $v_2$ such that $v_1 \triangledown v_2 \succeq v_1$ and also $v_1 \triangledown v_2 \succeq v_2$. E.g., $(0,5) \triangle (3,3) = (3,5)$.

For the example domains considered in this paper, all of these operators ($\ominus$, $\prec$, $\triangle$, and $\triangledown$) can be defined. If $\oplus$ is (element-wise) addition, then $\ominus$ is (element-wise) subtraction, and $\prec$ is defined like $\succ$ but with the comparison reversed. In this case, $\triangle$ and $\triangledown$ are defined as (element-wise) $\max$ and $\min$ respectively.

With help of the functions above, one can compute the *reverse Pareto front* of $\mathrm{ub}_v$ (the blue and red crosses in Fig. 3).

$$\mathrm{reverse}(\Theta) = \mathrm{nondom}(\{v_1 \triangle v_2 \mid v_1, v_2 \in \Theta\}) \tag{42}$$

And similarly, to reverse back:

$$\overline{\mathrm{reverse}}(\Theta) = \overline{\mathrm{nondom}}(\{v_1 \triangledown v_2 \mid v_1, v_2 \in \Theta\}) \tag{43}$$

The upper bounds $\mathrm{ub}_v$ can be combined using $\overline{\mathrm{nondom}}$:

$$\mathrm{ub}_2^* = \overline{\mathrm{nondom}} \left( \bigcup_{v \in \Theta_1} \mathrm{ub}_v \right) \tag{44}$$

However, the set $\mathrm{ub}_2^*$ is not always the best upper bound. E.g., in Fig. 3 it would not include $(7,1)$, and therefore would not consider the solution $(8,2)$ dominated, even though it is.

The quality of the upper bound can be improved by also considering the reverse Pareto fronts. The reverse Pareto fronts of the solutions $v \in \Theta$ can be combined using $\overline{\mathrm{nondom}}$ and then reversed back with $\overline{\mathrm{reverse}}$ as follows:

$$\mathrm{ub}_2' = \overline{\mathrm{reverse}} \left( \overline{\mathrm{nondom}} \left( \bigcup_{v \in \Theta_1} \mathrm{reverse}(\mathrm{ub}_v) \right) \right) \tag{45}$$

The final upper bound for the other subtree (the green circles in Fig. 3) is obtained by combining the two sets with $\overline{\mathrm{nondom}}$:

$$\mathrm{ub}_2 = \mathrm{ub} - \Theta_1 = \overline{\mathrm{nondom}}(\mathrm{ub}_2^* \cup \mathrm{ub}_2') \tag{46}$$

**Lower bounds** The set $\mathrm{lb}$ is a lower bound to $\Theta$, if for all solutions $v$ in $\Theta$ there is no solution in $\mathrm{lb}$ which is dominated by $v$:

$$\forall v \in \Theta : \neg \exists v' \in \mathrm{lb} : v \succ v' \tag{47}$$

We compute lower bounds similarly as Demirović and Stuckey [19]. We use the notation $\mathrm{LB}(s,d)$ for retrieving a lower bound from the cache or by computing it by using a similarity bound for a state $s = \langle \mathcal{D}, F \rangle$. Any optimal solution in the cache is also a lower bound. Furthermore, if a tree search node is infeasible (no feasible solution below the upper bound $\mathrm{ub}$ exists), then the upper bound for this node becomes a lower bound that is stored in the cache.

The lower bound may also be computed by using a *similarity bound*, but this only applies when the optimization task

1. has a subtraction operator $\ominus$;
2. does not have constraints;
3. is *context independent*; and
4. and the optimization task is *per-instance* (element-wise) additive.

**Definition B.1** (Context independent). An optimization task is *context independent* if and only if the cost function does not depend on any of the parent nodes' branching decisions.

For example, cost-sensitive classification with discounts, as introduced in this paper, is not context independent.

**Definition B.2** (Per-instance (element-wise) additive)**.** An optimization task is *per-instance (element-wise) additive*, if the cost function for label assignments can be written as the sum of the costs from a per-instance cost function $h$:

$$g(\langle \mathcal{D}, F \rangle, \hat{k}) = \sum_{(x,k) \in \mathcal{D}} h(x, k, \hat{k}) \tag{48}$$

If all these four conditions are met, the similarity bound can be used. Its definition is generalized from the definition given in [19]. Instead of subtracting the misclassification score per class from an existing solution, the worst-case value for instances of that class is subtracted by use of $\ominus$. Let $w_k$ be the worst case contribution of a single instance with class $k$ to the objective, and let $n_k$ be the number of instances in a dataset $\mathcal{D}$ that are not part of a dataset for a cached solution $\mathcal{D}^c$: i.e. if $\mathcal{D}_k = \{(x, k') \in \mathcal{D} \mid k' = k\}$, then $n_k = |D_k \setminus D_k^c|$. Now assume each of these $n_k$ instances is misclassified and therefore $w_k$ is added to the objective for each of these instances, then a lower bound can be computed from a cached solution $\Theta^c$ as follows:

$$\mathrm{lb}(\mathcal{D}) = \{v \ominus \sum_{k \in \mathcal{K}} n_k w_k \mid v \in \Theta^c\} \tag{49}$$

In this equation, the sum operator uses $\oplus$ to sum.

Once the lower bound for a search node is computed, it can be compared to the upper bound $\mathrm{ub}$ to check if any possible improvement exists, and if not, the node is pruned from the tree search. This happens whenever $\mathrm{lb} > \mathrm{ub}$, i.e., every element of the upper bound is reverse dominated by at least one element of the lower bound:

$$\mathrm{lb} > \mathrm{ub} \Leftrightarrow \forall v \in \mathrm{ub} : \exists v' \in \mathrm{lb} : v' \prec v \tag{50}$$

Computing these upper and lower bounds can be quite costly since it involves a product operation of two sets. To reduce the computation time for applying the subtraction or to combine two lower bounds for two subtrees, we apply a new procedure in which we first reduce the two sets to a representative subset using $\triangle$ in such a way that the resulting bounds are still valid. These representative subsets are then combined to get a new valid upper or lower bound. This decreases the computation time of the bound but prunes slightly fewer nodes.

**Sparse trees** Optimal decision trees may overfit on the training data [22] and therefore typically a sparsity coefficient is introduced that adds a cost for every branching node, as for example done in [10, 36, 52]. In the work presented here, we do not consider such a sparsity coefficient, although it could trivially be added. Instead, STreeD optimizes directly for a given maximum number of nodes. This number of nodes can then be tuned (as explained below). In the DP formulation in the main text, this is left out for brevity. For completeness, the DP formulation should be changed by considering the following equation in a branching node ($d > 0$), as is similarly proposed in [20]:

$$\mathrm{opt}\left(\bigcup_{f \in \mathcal{F}, i \in [0, n-1]} \mathrm{merge}\left(T(t(s, f), d-1, i), T(t(s, \bar{f}), d-1, n-i-1), s, f\right)\right) \tag{51}$$

With this addition, STreeD can be tuned to optimize the number of nodes to prevent overfitting. This is done by using part of the training data as validation data and train STreeD on the rest of the training data for an increasing number of nodes. This is repeated five times and the number of nodes that yields on average the best result is used as the final maximum number of nodes.

**Caching** STreeD supports two forms of caching: branch caching and dataset caching. Dataset caching is typically slightly faster than branch caching [20]. However, dataset caching cannot be used for context-dependent optimization tasks, because dataset similarity is no longer sufficient to identify similar subproblems. Similarly, for some optimization tasks, branch caching should be replaced by equivalent state caching, which we leave as future work. In this paper, dataset caching is used, except for cost-sensitive classification and group fairness, which are context dependent and use branch caching.

**Algorithm 1:** Tree search of depth $d$ for an optimization task $\langle g, t, \succ, \oplus, c, s_0 \rangle$ for state $s$, for a feature set $\mathcal{F}$ and a maximum number of nodes $n$.

---

$T(s, d, n, \text{ub})$

    **if** $d = 0 \vee n = 0$ **then**

        **return** $\text{opt}\left( \cup_{\hat{k} \in \mathcal{K}} g(s, \hat{k}), s, \text{ub} \right)$

    **if** $n > 2^d - 1$ **then**

        **return** $T(s, d, 2^d - 1, \text{ub})$

    **if** $d > n$ **then**

        **return** $T(s, d, d, \text{ub})$

    $\langle \Theta, \text{lb}, \text{stat} \rangle \leftarrow \text{cache}[s, d, n]$

    **if** $\text{lb} > \text{ub}$ **then return** $\emptyset$

    **if** $\text{stat} = \text{optimal}$ **then return** $\text{opt}(\Theta, s, \text{ub})$

    $\Theta \leftarrow \emptyset$

    **for** $f \in \mathcal{F}, n_L \in [0, n-1]$ **do**

        $n_R \leftarrow n - n_L - 1$

        $\text{lb}_L \leftarrow \text{LB}(t(s, \bar{f}), d - 1, n_L)$

        $\text{lb}_R \leftarrow \text{LB}(t(s, f), d - 1, n_R)$

        $\text{lb} \leftarrow \text{merge}(\text{lb}_L, \text{lb}_R, s, f)$

        **if** $\text{lb} > \text{ub}$ **then continue**

        $\text{ub}_L = \text{ub} \ominus g(s, f)$

        $\Theta_L \leftarrow T(t(s, \bar{f}), d - 1, n_L, \text{ub}_L)$

        **if** $\Theta_L = \emptyset$ **then continue**

        $\text{ub}_R = \text{ub} \ominus \Theta_L \ominus g(s, f)$

        $\Theta_R \leftarrow T(t(s, f), d - 1, n_R, \text{ub}_R)$

        **if** $\Theta_R = \emptyset$ **then continue**

        $\Theta_{\text{new}} \leftarrow \text{opt}(\text{merge}(\Theta_L, \Theta_R, s, f), s)$

        $\text{ub} \leftarrow \text{opt}(\text{ub} \cup \Theta_{\text{new}}, s)$

        $\Theta \leftarrow \text{opt}(\Theta \cup \Theta_{\text{new}}, s, \text{ub})$

    **if** $\Theta = \emptyset$ **then**

        $\text{cache}[s, d, n] \leftarrow \langle \emptyset, \text{ub}, \text{lower bound} \rangle$

        **return** $\emptyset$

    $\text{cache}[s, d, n] \leftarrow \langle \Theta, \Theta, \text{optimal} \rangle$

    **return** $\Theta$

---

**Pseudo-code** Pseudo-code for STreeD is given in Algorithm 1. If a leaf node is reached, the cost function is used to compute optimal solutions. The next two checks in the pseudo-code check if the maximum number of nodes and the maximum depth are compatible. If they are, the cache is checked for optimal solutions or a lower bound. The search is stopped if the cached lower bound is dominated by the current upper bound. If the cached solution is optimal, it is returned.

Then, for every possible branching feature and every possible node division, the algorithm does the following. It computes lower bounds for the left and right subtree, merges these lower bounds, and skips this feature if the upper bound dominates the lower bound. Then a recursive call is performed to compute optimal solutions to the left subtree. The upper bound for the left tree is updated by subtracting the branching costs. Solutions from this recursive call are used to update the upper bound for the right subtree and the right subtree is also solved. Solutions from both subtrees are merged into a new set of solutions which are used to update the current set of solutions and the upper bound.

When all branching features have been checked and no feasible solution is found (better than the upper bound), the current upper bound is stored as a lower bound. If feasible solutions have been found, the current set of solutions is stored in the cache and the solutions are returned.

In case the comparison operator $\succ$ is fully defined (a *total ordering* of all solutions exists), such as is the case with minimizing misclassification score, then the $\text{nondom}$ operation always returns a set of precisely one solution (or zero if infeasible). In this case, $\text{nondom}$ can be replaced by $\min$ (as defined by $\succ$), simplifying the implementation and computation.

**Depth-two solver**   Demirović et al. [20] have shown that the runtime of optimal decision tree search can be improved considerably by use of a special solver for trees of maximum depth two. The core idea is to pre-compute class frequencies by looping over only the *present features* for every instance in a dataset. We generalize this concept in STreeD and explain it below.

STreeD's use of the depth-two solver is restricted under the following conditions:

1. the cost of a leaf node can be expressed as a function over the contributions of individual instances;
2. the cost of a leaf node is *context independent* (see Def. B.1, note that the branching costs may be context dependent); and
3. the branching costs are equal for datasets with the same size, i.e., $|\mathcal{D}_1| = |\mathcal{D}_2| \rightarrow g(\langle \mathcal{D}_1, F \rangle, f) = g(\langle \mathcal{D}_2, F \rangle, f)$ for all $f \in \mathcal{F} \times \{0, 1\}$.

For all optimization tasks considered in this paper, these conditions are met.

To use the depth-two solver, the following functions and values need to be defined:

1. a 'depth-two' solution space $\mathcal{W}$;
2. an 'empty' initial solution value $w_0 \in \mathcal{W}$ (typically zero), the solution value of a leaf with zero instances;
3. a per-instance cost function $j : \mathcal{X} \times \mathcal{K} \times \mathcal{K} \rightarrow \mathcal{W}$ such that $j(x, k, \hat{k})$ returns the 'depth-two' costs of instance $(x, k)$ when assigned label $\hat{k}$;
4. a 'depth-two' combining operator $\oplus' : \mathcal{W} \times \mathcal{W} \rightarrow \mathcal{W}$ that combines two solution values into one;
5. a 'depth-two' subtract operator $\ominus' : \mathcal{W} \times \mathcal{W} \rightarrow \mathcal{W}$, such that $(w_1 \oplus' w_2) \ominus' w_2 = w_1$ for all $w_1, w_2 \in \mathcal{W}$;
6. a transformation function $q : \mathcal{W} \rightarrow \mathcal{V}$ that transforms a 'depth-two' solution value into a solution value in the original solution value space $\mathcal{V}$;
7. a branching cost function $r : \mathcal{S} \times \mathbb{N}^+ \times \mathcal{F} \times \{0, 1\} \rightarrow \mathcal{V}$, such that $r(s, n, f)$ returns the cost of branching on $f$ in state $s$ with $n = |\mathcal{D}|$.

In this paper, for all optimization tasks $\mathcal{W} = \mathcal{V}$, $\oplus' = \oplus$, and $j(x, k, \hat{k}) = g(\langle \{(x, k)\}, \emptyset \rangle, \hat{k})$, but this is not necessarily true for future optimization tasks.

The function $j$ provides a per-instance breakdown of the costs which are summed with the $\oplus'$ operator. These are used to precompute the costs of all possible leaf nodes. Let $n(f_i)$ and $n(f_i, f_j)$ be the size of $\mathcal{D}_{f_i}$ and $\mathcal{D}_{f_i, f_j}$ respectively. Similarly, let $w(\hat{k})$ be the cost of assigning all instances in $\mathcal{D}$ label $\hat{k}$, and let $w(\hat{k}, f_i)$ and $w(\hat{k}, f_i, f_j)$ be the cost of assigning all instances in $\mathcal{D}_{f_i}$ and $\mathcal{D}_{f_i, f_j}$ respectively. By generalizing the result from Demirović et al. [20], the following statements hold:

$$n(\bar{f}_i) = |\mathcal{D}| - n(f_i) \tag{52}$$

$$n(f_i, \bar{f}_j) = n(f_i) - n(f_i, f_j) \tag{53}$$

$$n(\bar{f}_i, f_j) = n(f_j) - n(f_i, f_j) \tag{54}$$

$$n(\bar{f}_i, \bar{f}_j) = |\mathcal{D}| - n(f_i) - n(f_j) + n(f_i, f_j) \tag{55}$$

$$w(\hat{k}, \bar{f}_i) = w(\hat{k}) \ominus' w(\hat{k}, f_i) \tag{56}$$

$$w(\hat{k}, f_i, \bar{f}_j) = w(\hat{k}, f_i) \ominus' w(\hat{k}, f_i, f_j) \tag{57}$$

$$w(\hat{k}, \bar{f}_i, f_j) = w(\hat{k}, f_j) \ominus' w(\hat{k}, f_i, f_j) \tag{58}$$

$$w(\hat{k}, \bar{f}_i, \bar{f}_j) = w(\hat{k}) \ominus' w(\hat{k}, f_i) \ominus' w(\hat{k}, f_j) \oplus' w(\hat{k}, f_i, f_j) \tag{59}$$

Furthermore, let $f \in x$ denote that feature $f$ is satisfied by $x$. The depth-two solver can now be generalized as shown in Algorithm 2. It first precomputes the dataset sizes and costs, then finds the best left and right solutions for each possible split, and finally returns the best solution.

## C   Cost-Sensitive Classification

For our cost-sensitive classification experiment, we consider a constant but asymmetric misclassification costs, test costs (feature costs) that are constant or may depend on previous selected tests. For

**Algorithm 2:** Tree search of depth two using a special depth-two solver.

---

$T_{d2}(s, \langle D, F \rangle)$

   // Pre-compute dataset sizes

   $n(f_i) \leftarrow 0 \quad \forall f_i \in \mathcal{F}$

   $n(f_i, f_j) \leftarrow 0 \quad \forall f_i, f_j \in \mathcal{F}$ s.t. $i < j$

   **for** $(x, k) \in \mathcal{D}$ **do**

      **for** $f_i \in x$ **do**

         $n(f_i) \leftarrow n(f_i) + 1$

         **for** $f_j \in x$ **do**

            $n(f_i, f_j) \leftarrow n(f_i, f_j) + 1$

   // Pre-compute costs

   $w(\hat{k}) \leftarrow w_0 \quad \forall \hat{k} \in \mathcal{K}$

   $w(\hat{k}, f_i) \leftarrow w_0 \quad \forall \hat{k} \in \mathcal{K}, f_i \in \mathcal{F}$

   $w(\hat{k}, f_i, f_j) \leftarrow w_0 \quad \forall \hat{k} \in \mathcal{K}, f_i, f_j \in \mathcal{F}$ s.t. $i < j$

   **for** $(x, k) \in \mathcal{D}, \hat{k} \in \mathcal{K}$ **do**

      $w(\hat{k}) \leftarrow w(\hat{k}) \oplus' j(x, k, \hat{k})$

      **for** $f_i \in x$ **do**

         $w(\hat{k}, f_i) \leftarrow w(\hat{k}, f_i) \oplus' j(x, k, \hat{k})$

         **for** $f_j \in x$ **do**

            $w(\hat{k}, f_i, f_j) \leftarrow w(\hat{k}, f_i, f_j) \oplus' j(x, k, \hat{k})$

   // Find optimal subtrees

   **for** $f_i, f_j \in \mathcal{F}$ s.t. $f_i \neq f_j$ **do**

      **for** $\hat{k}_L, \hat{k}_R \in \mathcal{K}$ s.t. $\hat{k}_L \neq \hat{k}_R$ **do**

         $\text{cost}_L \leftarrow q(w(\hat{k}_L, \bar{f}_i, \bar{f}_j)) \oplus q(w(\hat{k}_R, \bar{f}_i, f_j))$

         $\text{cost}_L \leftarrow \text{cost}_L \oplus r(t(s, \bar{f}_i), n(\bar{f}_i, \bar{f}_j), \bar{f}_j) \oplus r(t(s, \bar{f}_i), n(\bar{f}_i, f_j), f_j)$

         $\text{cost}_R \leftarrow q(w(\hat{k}_L, f_i, \bar{f}_j)) \oplus q(w(\hat{k}_R, f_i, f_j))$

         $\text{cost}_R \leftarrow \text{cost}_R \oplus r(t(s, f_i), n(f_i, \bar{f}_j), \bar{f}_j) \oplus r(t(s, f_i), n(f_i, f_j), f_j)$

         **if** $\text{cost}_L \succ \text{best-cost}_L(f_i)$ **then**

            $\text{best-cost}_L(f_i) \leftarrow \text{cost}_L$

         **if** $\text{cost}_R \succ \text{best-cost}_R(f_i)$ **then**

            $\text{best-cost}_R(f_i) \leftarrow \text{cost}_R$

   **return** $\text{opt}\left( \bigcup_{f \in \mathcal{F}} \text{merge}(\text{best-cost}_L(f), \text{best-cost}_R(f), s, f) \right)$

---

costs, we follow the naming conventions of [83]. Misclassification costs or test costs that depend on individual cases or on feature values are easy extensions and also fit in the STreeD framework, but are not considered here.

The following sections present the related work, the experiment setup and results, and a discussion of the results. The objective formulation is given in Eq. (9) in the main text.

### C.1 Related Work

Lomax and Vadera [55] provide an overview of methods for cost-sensitive classification with decision trees, most of which use top-down induction with varying cost functions for deciding on the best split [54, 68, 69, 80]. Each of these splitting criteria is mostly the same, except for a slight difference in the trade-off between accuracy and feature costs. This therefore is a good example of the drawback of top-down induction by splitting criteria: the objective is not directly translatable to a good splitting criterion. To address this issue Min and Zhu [61] generalize a number of those methods using a hyperparameter that decides on the trade-off between information gain and feature costs. ICET [82] does the same but uses a genetic algorithm to decide the hyperparameter. All of these methods except ICET, however, do not consider misclassification costs, and even ICET only does so indirectly.

Lomax and Vadera [56] co-optimize accuracy and costs by approaching the problem as a multi-armed bandit scenario. This means that in every branching node, they repeatedly try random splits (possibly

Table 1: Datasets used in cost-sensitive classification. $|\mathcal{F}_{disc}|$ is the number of categorical features in the dataset as preprocessed by [57]. $|\mathcal{F}_{bin}|$ is the number of resulting binary features.

| Dataset | $|\mathcal{D}|$ | $|\mathcal{F}_{disc}|$ | $|\mathcal{F}_{bin}|$ | $|\mathcal{K}|$ |
|---------|------|------|------|------|
| Annealing | 898 | 24 | 82 | 5 |
| Breast | 277 | 9 | 38 | 2 |
| Car | 1728 | 6 | 21 | 4 |
| Diabetes | 768 | 6 | 11 | 2 |
| Flare | 323 | 10 | 25 | 3 |
| Glass | 214 | 7 | 17 | 6 |
| Heart | 297 | 11 | 20 | 2 |
| Hepatitis | 154 | 16 | 20 | 2 |
| Iris | 150 | 4 | 12 | 3 |
| Krk | 28056 | 6 | 40 | 18 |
| Mushroom | 8124 | 21 | 111 | 2 |
| Nursery | 8703 | 8 | 26 | 5 |
| Soybean | 562 | 35 | 80 | 15 |
| Tictactoe | 958 | 9 | 27 | 2 |
| Wine | 178 | 13 | 32 | 3 |

multiple consecutive random splits) and choose the split which on average has the best expected costs based on this random exploration.

Maliah and Shani [58] formulate the problem as a Partially Observable Markov Decision Process (POMDP). However, they observe that the action and state space of their model is too big, and therefore they present a smaller MDP that approximates the POMDP. For the MDP model, they generate trees for all possible subsets of features using a standard heuristic that ignores costs. The MDP's state space is the set of all leaf nodes of all those generated trees. The resulting MDP is acyclic, and therefore its value function can be computed in one bottom-up pass through the belief states.

As far as we know, Nijssen and Fromont [66] and Zubek and Dietterich [94] are the only ones to provide an optimal formulation. The latter do so by formulating it as an MDP. However, in their experiments, they decide to use a relaxed formulation instead because of memory limitations.

## C.2 Experiment Setup

Table 1 lists the datasets used in this experiment [24, 41, 95]. These datasets were preprocessed and discretized according to the instructions in [57]. For methods that require binary features, the discretized features were binarized using one-hot encoding. Every dataset is split randomly 100 times in a train and test set, with 80% and 20% of the instances respectively.

We use the fixed feature costs as defined in [57]. The cost of misclassification strongly impacts the behavior of cost-sensitive algorithms. When misclassification costs are low, feature costs are comparatively high and branching is discouraged. Conversely, when misclassification costs are high, feature costs are comparatively low and therefore less important. In this case, overfitting also becomes more likely. Therefore, similar to the experiment setup in [57, 58] we vary the misclassification costs and generate misclassification matrices with low, middle, and high misclassification costs. These matrices are defined based on the feature costs and the class frequency as follows: First, calculate the maximum possible feature costs $C$ by summing the costs for all features. The default cost $C_{def}$ for low, middle, and high-cost experiments are defined as $1/6\ C$, $1/3\ C$, and $C$ respectively. By considering the relative frequency $f_k$ of class $k$, the misclassification costs $C_k$ for class $k$ are defined as follows:

$$C_k = \frac{C_{def}}{f_k |\mathcal{K}|} \tag{60}$$

For each run, a timeout of 300 seconds is used.

**Normalized costs** We report normalized costs, as defined in [82]. The normalized costs are calculated by dividing the obtained costs by the standard costs. As before, let $C$ be the sum of all feature costs, let $f_k$ be the relative frequency of class $k$, and let $C_{k,\hat{k}}$ be the misclassification cost when an instance with true label $k$ is assigned label $\hat{k}$. Then the standard costs is defined as follows:

$$C + \min_k \left(1 - f_k\right) \cdot \max_{k,\hat{k}} C_{k,\hat{k}} \tag{61}$$

These standard costs are lower than the maximum possible costs, but function as a good upper bound on the average costs [82].

In one aspect we deviate from the original definition. In the original definition, $f_k$ was calculated based on the whole dataset, but here it is based on the dataset that is examined (the train or test set).

### C.2.1 Methods

Apart from STreeD, the following methods are evaluated in our experiments:

**C4.5** [73] is an extension of the ID3 algorithm [72] and uses top-down induction (TDI) based on an information gain measure. After construction, a pruning mechanism is used to remove branching nodes that are likely to result in overfitting. C4.5 is cost-insensitive and therefore functions as a baseline heuristic.

**CSID3** [80] is a variant of ID3 with an updated splitting criterion. Instead of only information gain, it makes a trade-off between the information gain $I(f)$ and the cost $C(f)$ of feature $f$ by the splitting criterion:

$$\frac{I(f)^2}{C(f)}$$

CSID3 does not consider varying misclassification costs.

**EG2** [69] is also a variant of ID3 with an updated splitting criterion:

$$\frac{2^{I(f)} - 1}{(C(f) + 1)^\omega}$$

The parameter $\omega$, with values $0 \leq \omega \leq 1$, determines the weight of the cost factor. Following Turney [82], we set $\omega = 1$. EG2 also does not consider varying misclassification costs.

**IDX** [68] is yet another variant of ID3 with the splitting criterion:

$$\frac{I(f)}{C(f)}$$

The look-ahead procedure of IDX is not implemented. IDX also does not consider varying misclassification costs.

**MetaCost** [23] can be used to turn any error-based classifier into a cost-sensitive classifier. It iteratively fits a model, computes class probabilities per sample, and updates the training input labels such that the expected costs are minimized based on the class probabilities. Note that MetaCost only considers misclassification costs and ignores feature costs. We run MetaCost with 10 iterations.

**TMDP** [58] formulates the problem as a Markov Decision Process (MDP). For every possible subset of features, it uses C4.5 to generate a normal cost-insensitive tree, and every leaf from the resulting tree is added as a state to the MDP. Each state is described by the path to the leaf. The state with the empty path is the starting state. In every state, the possible actions are to measure a feature or to classify a sample. If a feature is measured, the next state is the one described by the path that results from appending the measured feature to the current path. When classifying, the MDP moves to the terminal state. This MDP therefore is acyclic, and the values of the states can be computed in reverse topological order by using the training data. Based on these values, the tree can now be constructed by choosing the action with maximum value, beginning in the start node, and splitting into multiple nodes when selecting a feature to measure. This is repeated until all nodes are leaf nodes.

Because TMDP creates trees for every possible subset of the feature set, the total number of possible states explodes for an increasing number of features. Therefore, the authors

Table 2: Out-of-sample normalized costs (%). Significantly ($p < 5\%$) best results are marked in bold. Timeouts are marked as '-'.

| Dataset | C4.5 | CSID3 | EG2 | IDX | MetaCost | TMDP | STreeD |
|---|---|---|---|---|---|---|---|
| Annealing | 5.6 | 5.6 | 2.2 | 2.2 | 5.7 | 1.6 | **1.2** |
| Breast | 43.9 | 43.4 | 41.0 | 40.7 | 54.8 | **21.0** | **20.4** |
| Car | 43.1 | 42.2 | 39.8 | 39.9 | 42.2 | 16.1 | **14.8** |
| Diabetes | 50.2 | 44.5 | 43.4 | 43.4 | 55.9 | **14.2** | **14.1** |
| Flare | 17.9 | **15.7** | **15.4** | **15.3** | 21.8 | 18.0 | 16.7 |
| Glass | 37.7 | 35.6 | 30.1 | 29.0 | 39.4 | **14.3** | **14.4** |
| Heart | 45.4 | 23.0 | 20.0 | 19.9 | 44.7 | **9.5** | **9.3** |
| Hepatitis | 32.2 | 24.3 | 20.0 | 19.7 | 32.2 | 16.2 | **15.6** |
| Iris | 38.8 | 34.0 | 29.5 | 29.7 | 40.9 | **12.8** | **12.8** |
| Krk | 6.5 | 6.2 | 6.3 | 6.3 | 6.5 | - | **1.7** |
| Mushroom | 16.4 | 8.3 | 5.6 | 4.5 | 16.4 | - | **1.9** |
| Nursery | 0.4 | 0.4 | 0.4 | 0.4 | 0.4 | 0.1 | **0.1** |
| Soybean | 12.8 | 16.3 | 19.8 | 21.1 | 12.6 | 15.4 | **11.4** |
| Tictactoe | 45.0 | 45.0 | 45.0 | 45.0 | 45.0 | **16.1** | **16.1** |
| Wine | 25.0 | 18.8 | 18.1 | 17.5 | 24.8 | 12.2 | **10.4** |

propose that for a larger number of features, TMDP only tests all possible subsets of the 10 most costly features together with all the cheap features considered by default. In the analysis of TMDP here, the same approach is used.

Note that TMDP is the only method that is evaluated here that considers multi-way splits instead of binary splits.

For these methods, we use an implementation that is available online.[2] C4.5 is implemented in Weka, with CSID3, EG2, and IDX implemented as C4.5 but with the new splitting criterion. Leaf nodes do not select the majority class, but the class with the lowest total misclassification costs.

## C.3 Results

**Normalized costs**  Table 2 shows the normalized cost results for all methods on a variety of datasets. On 14 out of 15 datasets STreeD has the best performance or is not statistically significantly worse than the method with the best performance. The results show that TMDP and STreeD have similar performance and significantly outperform the other heuristics on all datasets except Flare. Therefore, the rest of the analysis will focus on TMDP and STreeD.

Table 3 shows the cost results for TMDP and STreeD for low, middle, and high misclassification costs. The difference in the results can be explained by several factors: multi-way splits versus binary splits, and the quality of the binarization. TMDP considers multi-way splits and therefore has a larger solution space, but does not necessarily find the best tree in this solution space according to the training data. STreeD, on the other hand, only considers binary decision trees and its solution quality depends on the binarization. However, it always finds the best binary decision tree for the training data.

**Runtime**  Figure 4 shows the difference in runtime for STreeD (hypertuned with $d = 4$) and TMDP. As can be seen, STreeD is often an order of magnitude faster than TMDP, and is significantly faster than TMDP for all datasets ($p < 5\%$), except for Iris, the smallest dataset.

**Interpretability**  Table 4 shows how the methods compare in terms of interpretability metrics. When only looking at the numbers, TMDP has the best score, and STreeD follows shortly after. The heuristics, as expected, score much worse. However, TMDP allows for multi-way split trees and the others do not. According to Piltaver et al. [71], a higher branching factor increases the classifying time of an average user, although this effect can only be seen with branching factors higher than 3.

---

[2]https://github.com/shanigu/CostSensitive

Table 3: Test normalized cost (%) for varying misclassification costs. Significantly ($p < 5\%$) best results are marked in bold. Timeouts are marked as '-'.

|  | Low costs | | Middle costs | | High costs | |
|---|---|---|---|---|---|---|
| Dataset | STreeD | TMDP | STreeD | TMDP | STreeD | TMDP |
| Annealing | **1.3** | 1.6 | **1.2** | 1.6 | **1.2** | 1.5 |
| Breast | 8.3 | 8.3 | 15.7 | 15.6 | **37.3** | 39.1 |
| Car | 9.8 | 9.8 | **14.0** | 15.8 | **20.5** | 22.9 |
| Diabetes | 7.3 | 7.2 | **11.6** | 11.8 | 23.4 | 23.5 |
| Flare | 9.9 | 9.8 | **15.2** | 16.3 | **24.9** | 28.0 |
| Glass | 10.9 | 10.8 | 13.8 | 14.1 | 18.5 | 18.1 |
| Heart | 4.0 | 4.0 | 7.2 | 7.5 | 16.7 | 17.1 |
| Hepatitis | **8.4** | 9.2 | 12.9 | 13.5 | 25.6 | 26.0 |
| Iris | 7.5 | 7.5 | 10.7 | 10.7 | 20.2 | 20.3 |
| Krk | **1.8** | - | **1.8** | - | **1.7** | - |
| Mushroom | **1.7** | - | **1.8** | - | **2.1** | - |
| Nursery | 0.1 | 0.1 | 0.1 | 0.1 | 0.1 | 0.1 |
| Soybean | **10.9** | 18.7 | **11.6** | 16.3 | 11.8 | **11.3** |
| Tictactoe | 6.8 | 6.8 | 12.7 | 12.7 | 28.8 | 28.9 |
| Wine | **8.0** | 11.3 | **11.9** | 12.5 | **11.2** | 12.8 |

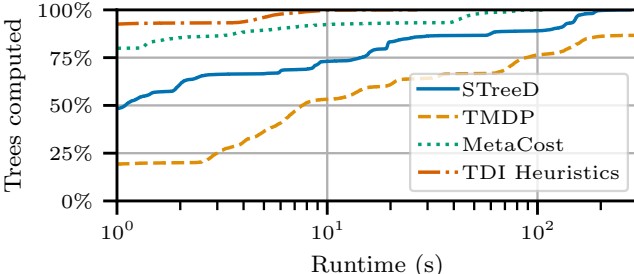

Figure 4: Runtime (s) comparison. The top-down induction heuristics (C4.5, CSID3, EG2, and IDX) have been grouped together as 'TDI Heuristics' because they almost do not differ in runtime. A timeout of 300 seconds is used. Note the logarithmic x-axis.

Table 4: Interpretability metrics. Significantly ($p < 5\%$) best results are marked in bold. Datasets for which TMDP exceeded the timeout limit are ignored.

|  | C4.5 | CSID3 | EG2 | IDX | MetaCost | TMDP | STreeD |
|---|---|---|---|---|---|---|---|
| Tree depth | 7.9 | 7.1 | 7.1 | 7.1 | 7.2 | **1.9** | 2.2 |
| Leaf nodes | 26.1 | 27.2 | 28.6 | 28.7 | 24.8 | 7.4 | **5.4** |
| Question length | 4.4 | 4.7 | 5.0 | 5.0 | 4.7 | **1.7** | 2.8 |

Therefore, we can conclude that both TMDP and STreeD clearly outperform the heuristics in terms of interpretability.

## C.4 Discussion

The results above clearly show the advantage of TMDP and STreeD over the heuristics. For most datasets, the cost performance of TMDP and STreeD was much better than the heuristics. This is best explained by the fact that the splitting criteria of the heuristics do not directly capture the objective, and in these cases even ignore the misclassification costs.

Comparing TMDP and STreeD is more difficult. In terms of costs, both at times perform best, although on average STreeD's out-of-sample costs are (significantly) lower: $13.3\%$ normalized costs versus TMDP's $14.3\%$. Both methods show some overfitting and this could be further addressed. Direct comparison is difficult though, because they differ in many ways: TMDP has categorical data

with multi-way splits and seeks a tree without a depth limit. STreeD has binary data with binary splits and seeks the best tree up to some maximum size.

The runtime of STreeD is at least an order of magnitude lower than TMDP. One of the consequences here is that TMDP was not able to find good trees for several datasets within the given time limit. Moreover, these results are produced with TMDP's limit of considering only subsets for the 10 most costly features. Without this limit, TMDP's runtime would have been worse. STreeD therefore is more scalable than TMDP.

Both STreeD and TMDP have good interpretability and clearly outperform the heuristics. The trees returned by the heuristics are often too large to be considered interpretable.

From these results it can be concluded that STreeD can successfully be applied to cost-sensitive classification, outperforming the state-of-the-art in cost performance and runtime, while providing small interpretable trees.

# D    Prescriptive Policy Generation

Decision trees can also be used to prescribe policies. Given a set of instances, actions, and outcomes, the decision tree prescribes which action for an unseen instance will give the best predicted outcome. An example use case is prescribing medical treatment on basis of historical treatment response.

However, since historical data does not necessarily have the outcome for every possible action, the prescriptive model needs to reason about the counterfactuals. Therefore, conceptually, this problem can be split into a prediction and a prescription problem. The prediction problem predicts the counterfactuals. The prescription problem decides what action to take.

The following sections present first the related work and then a formal problem definition and other preliminaries. We then describe the experiments and conclude with a discussion.

## D.1    Related Work

Athey and Imbens [7] use trees to estimate the causal effect of treatments. They use part of the dataset to estimate heterogeneous partitions and the rest is used to estimate treatment effects in the leaves, including confidence intervals for these values. However, this method does not directly assign policies, and in the case of non-binary treatments, several trees need to be combined to produce a policy tree, reducing its interpretability.

Kallus [45] combines the prediction and prescription into one learning problem and proposes both a top-down induction heuristic and an optimal MIP method that directly optimizes prescription error. Their methods work under the assumption that the training data is either produced in a randomized experiment or that the generated tree partitions the data such that the historical treatment assignment becomes independent of the individual. However, in practice, the optimal MIP method does not scale to sufficient depth to reach such a partition. Their results therefore also show that the optimal MIP method in most situations performs worse than other solution methods.

Bertsimas et al. [12] improve on [45] by observing that the performance of prescriptive trees relies not only on a good heterogeneous partition but also on the quality of the prediction of outcomes. Therefore they propose an optimal MIP formulation that co-optimizes both the prescription and prediction error in one objective. To circumvent the scalability issues for finding the global optimum, they repeatedly use a coordinate descent method [11] to find local optima and select the best tree.

Biggs et al. [13] and Zhou et al. [92] note the disadvantage of trying to incorporate the predictive aspect into the MIP model. Instead they use a separate predictive teacher model and use this model to guide a top-down greedy heuristic. Zhou et al. [92] also provide an optimal recursive search method, but they did not incorporate any of the scalability improvements from similar approaches [3, 20] such as caching and lower-bounding.

Amram et al. [5] take a similar approach, but instead of a greedy heuristic, they propose to adapt the MIP model of [12] by updating the objective to incorporate the counterfactual data obtained from a predictive model, using objectives from causal inference. Their experiments show its advantage over both the greedy heuristics and the method by Bertsimas et al. [12].

Jo et al. [43] analytically show the limitations of both [45] and [12]: their assumption is generally not satisfied in conditionally randomized experiments. Both methods can return trees that actively violate their own assumption. Therefore Jo et al. [43], like Amram et al. [5], propose to drop this assumption and use separate predictive and prescriptive models. Apart from this, they show that under mild conditions their method provides an optimal out-of-sample policy when the number of training samples approaches infinity. Their model also allows for the addition of budget constraints. They propose to use the same objectives as Amram et al. [5], but their MIP formulation is based on the more scalable formulation of Aghaei et al. [2], so unlike Bertsimas et al. [12] and Amram et al. [5] they do not need to fall back on coordinate descent for local optima. However, despite their scalability improvements, almost half of the instances in their experiments are still not solvable within one hour.

Finally, Subramanian et al. [79] find near-optimal solutions by using column generation to improve scalability. Furthermore, their model allows for the addition of a variety of constraints.

### D.2 Preliminaries

We consider a dataset $\mathcal{D}$ containing instances $(x, k, y)$ with $x \in \mathcal{X}$ the feature vector, $k \in \mathcal{K}$ the label or historically assigned treatment, and $y \in \mathbb{R}$ the observed outcome value. We assume no information on the historical treatment assignment policy. Let $Y(x, k)$ be the value outcome when assigning treatment $k$ to an individual described by $x$. The goal now is to find a policy $\pi : \mathcal{X} \to \mathcal{K}$ that maximizes the expected outcome $Q(\pi) = \mathbb{E}[Y(x, \pi(x))]$.

Because not every treatment was applied to each person, we do not have full information on the values of $Y(x, k)$, and therefore we need a teacher model that provides information on the counterfactuals, based on the data that we do have. Similar to [5, 43], we will here consider three teacher models: the direct method (DM), inverse propensity weighting (IPW), and the doubly robust approach (DR).

**Direct method**  The direct method (also called Regress and Compare) trains a teacher model $\hat{v}_k$ for the expected outcome $y$ for each possible treatment $k$ by splitting the training data based on the historical treatment. The objective value $Q$ can therefore be formulated as follows:

$$Q_{\text{DM}}(\mathcal{D}, \pi) = \frac{1}{|\mathcal{D}|} \sum_{(x,k,y) \in \mathcal{D}} \hat{v}_{\pi(x)}(x) \tag{62}$$

**Inverse propensity weighting**  The propensity scores $\mu(x, k) = \mathbb{P}(k \mid x)$ give the probability of a treatment $k$ for a given feature vector $x$. IPW implicitly models the counterfactuals by reweighing the actual data based on the inverse of their propensity scores. This requires the training of a propensity score model $\hat{\mu}(x, k)$. Any ML model can be used to learn $\hat{\mu}$. Given a model for $\hat{\mu}$, the objective value $Q$ of the policy becomes:

$$Q_{\text{IPW}}(\mathcal{D}, \pi) = \frac{1}{|\mathcal{D}|} \sum_{(x,k,y) \in \mathcal{D}} \frac{\mathbb{1}(\pi(x) = k)}{\hat{\mu}(x, k)} \tag{63}$$

**Doubly robust**  Finally, the doubly robust method [25] combines IPW and DM as follows:

$$Q_{\text{DR}}(\mathcal{D}, \pi) = \frac{1}{|\mathcal{D}|} \sum_{(x,k,y) \in \mathcal{D}} \left( \hat{v}_{\pi(x)}(x) + (y - \hat{v}_k(x)) \frac{\mathbb{1}(\pi(x) = k)}{\hat{\mu}(x, k)} \right) \tag{64}$$

### D.3 Objective Task Formulation

Once the objective value function $Q$ is known as stated above, this can be directly applied in our framework since each of the objectives $Q_{\text{IPW}}$, $Q_{\text{DM}}$ and $Q_{\text{DR}}$ are separable, once the teacher model values resulting from $\hat{\mu}$ and $\hat{v}$ have been precomputed. For each of these functions, the combining operator is addition, the comparator is $>$ (maximization) and the cost function is determined by the value of the policy $\pi_k$ that assigns treatment $k$ to all individuals:

$$g(\mathcal{D}, k) = Q(\mathcal{D}, \pi_k) \tag{65}$$

Table 5: Mean effect and treatment effect functions for the synthetic dataset, based on [7].

| | $\phi(x)$ | $\kappa(x)$ |
|---|---|---|
| $f = 2$ | $\frac{1}{2}x_1 + x_2$ | $\frac{1}{2}x_1$ |
| $f = 10$ | $\frac{1}{2}\sum_{j=1}^{2} x_j + \sum_{j=3}^{6} x_j$ | $\sum_{j=1}^{2} \max(0, x_j)$ |
| $f = 20$ | $\frac{1}{2}\sum_{j=1}^{4} x_j + \sum_{j=5}^{8} x_j$ | $\sum_{j=1}^{4} \max(0, x_j)$ |

## D.4 Experiments

The following experiments compare the performance of STreeD with the state-of-the-art optimal methods for prescriptive policy generation. In this analysis, our main consideration is runtime performance, since the other methods optimize precisely the same objective. We examine the methods on synthetic data and on a real-life dataset from the literature.

**Jo-PPG-MIP** [43] is a MIP model based on a max-flow formulation for optimal decision trees from [2] but with the aforementioned $Q$-function as the objective. We call this method Jo-PPG-MIP, after their first author. Their experiments show their advantage over previous MIP methods [12] and [45]. Our implementation of Jo-PPG-MIP is based on the open-source code for [2], for which we updated the objective.[3]

**CAIPWL-opt** [92] also observe the scalability issues with MIP-based methods and therefore propose a recursive tree-search algorithm that maximizes the doubly robust metric. Their method does not use caching, and therefore this method does not qualify as a dynamic programming approach. They suggest their optimal method is best for trees of at most depth three. For deeper trees, they suggest using approximate methods. In our experiments we use their publicly available R-package.[4]

### D.4.1 Synthetic Data

For the experiment with synthetic data, we follow the setup as done in [7, 43]. We consider two possible treatments $K = \{0, 1\}$, input vectors $X$, which consist of $f$ independent variables with distribution $N(0, 1)$ and for each $X_i$ corresponding potential outcomes $Y_i(k)$:

$$Y_i(k) = \phi(X_i) + \frac{1}{2}(2k - 1) \cdot \kappa(X_i) + \varepsilon_i. \tag{66}$$

In this equation, $\phi(x)$ is the mean effect, $\kappa(x)$ is the treatment effect, and $\varepsilon_i \sim N(0, 0.1)$ is independent noise. Based on [7] we consider three variations of $\phi$ and $\kappa$, for $f = 2, 10, 20$, as shown in Table 5. As can be seen from this table, for $f = 2$, the mean effect $\phi$ depends on both variables, and the treatment effect only on one. For $f = 10$ and $f = 20$, the mean effect depends on a subset of the variables. The treatment effect also depends on a subset of the variables, but only when those values are positive. Moreover, for these two cases, some variables are not related to the potential outcomes at all and function as noise in the dataset. For the training dataset, per instance, we then select the optimal treatment (the treatment with the maximum value for $Y_i$) as the historical treatment with probability $p \in \{0.1, 0.25, 0.5, 0.75, 0.9\}$ and store the corresponding outcome in the dataset. We repeat this process five times and generate training sets and test sets with sizes 500 and 10000 respectively.

We binarize each of the original features into 10 bins with a similar frequency count. The teacher models are then trained on the original data. For IPW we train a decision tree model (dt, computed with CART), logistic regression (log), and the true propensity scores. For DM we train a linear regression (lr) and a lasso regression with $\alpha = 0.08$. For DR we use all six combinations of the IPW and DM methods.

---

[3] https://github.com/pashew94/StrongTree
[4] https://github.com/grf-labs/policytree

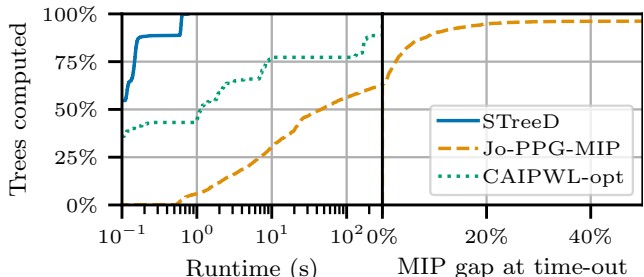

Figure 5: Proportion of instances that can be solved per method for the synthetic datasets within a given runtime (s). Note the logarithmic x-axis for runtime.

Table 6: Results for the Warfarin dataset. The reported runtime is the average of runs that did not result in a timeout (300s).

| | Jo | | CAIPWL-opt | | STreeD | OOS Correct |
|---|---|---|---|---|---|---|
| $d$ | Timeout | Time (s) | Timeout | Time (s) | Time (s) | Treatment |
| 1 | - | 30 | - | < 1 | < 1 | 83.1% |
| 2 | 62% | 215 | - | 5 | < 1 | 84.6% |
| 3 | 100% | - | - | 96 | < 1 | 86.6% |
| 4 | 100% | - | 100% | - | < 1 | 88.0% |
| 5 | 100% | - | 100% | - | 3 | 89.7% |

**Runtime** The setup above results in 75 scenarios that are solved with 11 different teacher methods. We ran all three methods with a maximum tree depth of 1, 2, and 3, resulting in 2475 problems for each method to solve.

Figure 5 shows the runtime performance by listing how many instances of the synthetic dataset can be solved within the specified amount of runtime. Note that this is only the time required to build the tree. It does not include the time to train the teacher model. These results show that STreeD is several orders of magnitude faster than Jo-PPG-MIP and also scales much better than CAIPWL-opt while providing exactly the same solutions.

### D.4.2 Warfarin Dosing

Similar to previous studies, we test our method on the Warfarin dosing use-case [12, 43, 45]. Warfarin is an anticoagulant, but the appropriate dose may vary up to a factor 10 among patients. Based on experimental tests, a dataset is compiled which prescribes for 5700 patients a suggested dose [39]. For our experiment, unless otherwise specified, we follow the setup by [43].

We calculate the square root of the suggested weekly dose by the formula presented in the supplementary material section S1e of [39] and add noise of $N(0, 0.02)$. This number is then squared and divided by seven to give the suggested daily dose. The features used in their formula are also the features used for the dataset, preprocessed as specified by them. The only three non-binary features (age, weight, and height) are binarized by splitting them into five buckets with approximately the same number of instances. Instances with missing values for weight, height, age, and the therapeutic dose of Warfarin are removed. This leaves 4490 instances. The daily suggested dose is discretized into three groups, with $k^{opt} = 0$ if the dose is less than 3 mg/day, $k^{opt} = 1$ if the dose is between 3 and 7 mg/day, and $k^{opt} = 2$ if the dose is larger than 7 mg/day. The observed outcome is $Y_i(k_i) = 1$, if $k_i = k_i^{opt}$; and $Y_i(k_i) = 0$ otherwise.

For simulating historical data, three different setups are tested. In the first, every individual receives a uniformly random treatment $k_i$ in the historical data with the corresponding outcome $Y_i$, to simulate a marginally randomized setting. In the second and third setups, historical treatments are obtained by using the formula for the weekly dose from [39] mentioned above with each coefficient in this formula perturbed by a random amount from $U(-0.06, 0.06)$ and $U(-0.11, 0.11)$ respectively, to change the probability of the historical data having the correct assignment. Each setup is used five

Table 7: Runtime (s) for minimizing biobjective misclassification score for two classes, with timeouts beyond 600s denoted with '-'. Best results are shown in bold. Average of five runs.

| Dataset | $\|\mathcal{D}\|$ | $\|\mathcal{D}^+\|$ | $\|\mathcal{D}^-\|$ | $\|\mathcal{F}\|$ | $d = 3$ | | $d = 4$ | |
|---|---|---|---|---|---|---|---|---|
| | | | | | MurTree | STreeD | MurTree | STreeD |
| Anneal | 812 | 625 | 187 | 93 | < 1 | **< 1** | 24 | **2** |
| Audiology | 216 | 57 | 159 | 148 | < 1 | **< 1** | 54 | **2** |
| Australian-Credit | 653 | 357 | 296 | 125 | 2 | **< 1** | 139 | **19** |
| Breast-Wisconsin | 683 | 444 | 239 | 120 | < 1 | **< 1** | 61 | **7** |
| COMPAS | 6907 | 3196 | 3711 | 12 | 2 | **< 1** | 7 | **< 1** |
| Diabetes | 768 | 500 | 268 | 112 | 3 | **< 1** | 117 | **13** |
| Fico | 10459 | 5000 | 5459 | 17 | 6 | **< 1** | 26 | **3** |
| German-Credit | 1000 | 700 | 300 | 112 | 6 | **< 1** | 164 | **34** |
| Heart-Cleveland | 296 | 160 | 136 | 95 | < 1 | **< 1** | 43 | **6** |
| Hepatitis | 137 | 111 | 26 | 68 | < 1 | **< 1** | 6 | **< 1** |
| Hypothyroid | 3247 | 2970 | 277 | 88 | < 1 | **< 1** | 28 | **3** |
| Ionosphere | 351 | 225 | 126 | 445 | 26 | **11** | - | - |
| Kr-vs-kp | 3196 | 1669 | 1527 | 73 | < 1 | **< 1** | 22 | **3** |
| Letter | 20000 | 813 | 19187 | 224 | 46 | **7** | - | **469** |
| Lymph | 148 | 81 | 67 | 68 | < 1 | **< 1** | 8 | **1** |
| Mushroom | 8124 | 4208 | 3916 | 119 | < 1 | **< 1** | **< 1** | < 1 |
| Pendigits | 7494 | 780 | 6714 | 216 | 13 | **4** | - | **240** |
| Primary-Tumor | 336 | 82 | 254 | 31 | < 1 | **< 1** | < 1 | < 1 |
| Segment | 2310 | 330 | 1980 | 235 | < 1 | < 1 | **< 1** | < 1 |
| Soybean | 630 | 92 | 538 | 50 | < 1 | **< 1** | 3 | **< 1** |
| Splice-1 | 3190 | 1655 | 1535 | 287 | 17 | **9** | - | - |
| Tic-Tac-Toe | 958 | 626 | 332 | 27 | < 1 | **< 1** | 1 | **< 1** |
| Vehicle | 846 | 218 | 628 | 252 | 5 | **2** | - | **130** |
| Vote | 435 | 267 | 168 | 48 | < 1 | **< 1** | 3 | **1** |
| Yeast | 1484 | 463 | 1021 | 89 | 5 | **< 1** | 68 | **8** |

times to generate historical treatments. Each of these 15 datasets is then randomly split five times into a training and test set with 75% and 25% of the instances respectively. For each of these, a decision tree is trained for IPW scores and a random forest regressor for DM estimates.

**Results**   Table 6 shows the results for applying Jo-PPG-MIP and STreeD to the Warfarin case. The results are very similar for the different teacher methods and the probability of having the correct assignment in the historical data.[5] Therefore we just show the average results for trees of depth 1-5.

These results show that STreeD is several orders of magnitude faster than both Jo-PPG-MIP and CAIPWL-opt. Because of this, STreeD can find optimal trees for larger depth, which for this problem results in a better out-of-sample (OOS) correct treatment rate.

## E   Nonlinear Classification Metrics

For imbalanced datasets, it is beneficial to use metrics other than accuracy, such as F1-score and Matthews correlation coefficient. However, these metrics are nonlinear and can therefore not (easily) be optimized with state-of-the-art methods for optimal decision trees. To address this, Demirović and Stuckey [19] developed a DP-based approach for binary classification that can optimize any function that is monotonic with respect to the misclassification score for both classes. This class of functions includes F1-score and Matthews-correlation coefficient.

STreeD can also optimize this whole class of functions by using the fact that multiple separable optimization tasks can be combined to form a new separable optimization task, according to Prop. A.4.

---

[5]This is different from the results in [43]. We have corresponded with the authors about this but were not able to reproduce their results. We will include the test setup in our repository to enable reproducibility of the results.

The next section provides more related work. Then we show experimental results for applying STreeD to nonlinear objectives.

## E.1 Related Work

Lin et al. [52] also propose a DP method for solving a variety of objectives, such as balanced accuracy and F1-score. However, they observe that F1-score is harder to optimize than other metrics that are monotonic in relation to the number of false positives and negatives because the best assignment in one leaf node is dependent on the assignment in another node. Therefore, they simplify the problem by introducing a parameter $\omega$ that helps determine which label should be assigned in leaf nodes, but as a result, the solution is no longer guaranteed to be optimal.

Xin et al. [91] present a DP method for finding the full Rashomon set of sparse decision trees, which is the set of almost-optimal decision trees (including the optimal). They then show how a Rashomon set for trees optimized for accuracy can be used to find the Rashomon set for F1-score. For this, they provide a bound on the accuracy score such that an Accuracy Rashomon set covers the F1-score Rashomon set.

## E.2 Experiments

In our experiments, we evaluate STreeD's performance on finding trees that maximize F1-score. F1-score is the harmonic mean of precision and recall and can be calculated as follows based on the number of true positives ($tp$), false positives ($fp$), and false negatives ($fn$): $tp/(tp + 0.5(fp + fn))$.

**Setup**   We compare STreeD with the MurTree algorithm from [19] on a set of binarized datasets with binary class as described in Table 7. The datasets are originally from the UCI repository [24] and from [6, 30, 41, 59, 81, 96, 97]. We use the binarized datasets from the MurTree repository.[6] For each dataset, both algorithms need to find a tree of depth three and four that optimizes F1-score. We repeat the tests five times and report the average runtime.

**Results**   Table 7 shows how STreeD compares to MurTree on optimizing nonlinear metrics. By using the geometric mean of the relative performance [31], we can see that for trees of depth three, STreeD is on average 7.1 times faster than MurTree; for trees of depth four, it is on average 6.8 times faster. This performance difference can be partly explained by the new upper and lower bound technique, as explained in Appendix B.

This same method could also be used to optimize other nonlinear metrics that are monotonic with respect to the false positive and negative scores, such as Matthews correlation coefficient.

## F   Group Fairness

In many cases, removing sensitive features is not sufficient to prevent discrimination [28, 70], and therefore fairness constraints are imposed on ML models. We here consider two versions of *group fairness*: demographic parity and equality of opportunity [16, 60]. We leave individual fairness notions as future work.

Group fairness metrics impose equality on groups defined by some sensitive feature. Demographic parity requires equal probability of a positive result for each group, and equality of opportunity requires equal true positive rate for each group. Both of these can be optimized using STreeD by adding the threshold constraints as defined in Eq. (11) for demographic parity and Eq. (71) for equality of opportunity, as introduced below.

The next section provides more related work on incorporating fairness constraints in decision tree search. We then derive how threshold constraints can be used to model fairness constraints in STreeD. The final section shows experimental results for imposing fairness constraints in STreeD.

---

[6] `https://bitbucket.org/EmirD/MurTree-bi-objective`

## F.1 Related Work

Fairness in decision trees was first considered by Kamiran and Calders [46] who proposed to preprocess the data to guide the decision tree training process to less discriminating trees. Later, Kamiran et al. [47] proposed new splitting criteria to enable top-down induction using a mix of information gain in the outcome label and the sensitive feature. However, they concluded that this was not sufficient and instead proposed after-the-fact relabeling to get good results.

Fairness constraints were first considered in MIP models for optimal decision trees by Verwer and Zhang [85]. Aghaei et al. [1] later developed a MIP model for optimal trees for either a group fairness or individual fairness constraint. Jo et al. [44] propose a new MIP model based on a max-flow formulation that scales better than previous models.

These MIP models, however, still do not scale well beyond small datasets and small tree depths. Therefore Wang et al. [90] propose to break down the problem and use MIP as a look-ahead procedure that decides on the best feature to split on, but does not solve the whole problem as a monolith. Their experiments show better scalability, but their method is no longer guaranteed to return optimal solutions.

Van der Linden et al. [53], on the other hand, propose an optimal DP method that optimizes a global fairness constraint by considering upper and lower bounds on the final discrimination value for subtrees to enable early comparison and pruning. Their results show order-of-magnitude improvements on runtime, and their method can handle much larger datasets than both the MIP methods and the iterative approach.

## F.2 Optimization Task Formulation

In this section we derive threshold constraints for demographic parity and equality of opportunity.

**Demographic parity**  As stated in the main text, demographic parity means that the probability of the positive outcome is the same for people of different groups:

$$P(\hat{y} = 1 \mid a = 1) = P(\hat{y} = 1 \mid a = 0) \tag{67}$$

Typically, this is simplified to limiting the difference between the probabilities to some small percentage $\delta$. Let $\hat{N}(y, a)$ be the number of group $a$ receiving the positive outcome, and $\hat{N}(\bar{y}, a)$ the number of group $a$ receiving the negative outcome, etc. Then Eq. (67) becomes:

$$\left| \frac{\hat{N}(y, a)}{N(a)} - \frac{\hat{N}(y, \bar{a})}{N(\bar{a})} \right| \le \delta \tag{68}$$

By observing that $\hat{N}(\bar{y}, a)/N(a) = 1 - \hat{N}(y, a)/N(a)$, and $\hat{N}(\bar{y}, \bar{a})/N(\bar{a}) = 1 - \hat{N}(y, \bar{a})/N(\bar{a})$, Eq. (68) can be rewritten to the two constraints shown in the main text:

$$\frac{\hat{N}(y, a)}{N(a)} + \frac{\hat{N}(\bar{y}, \bar{a})}{N(\bar{a})} \le 1 + \delta \qquad \qquad \frac{\hat{N}(y, \bar{a})}{N(\bar{a})} + \frac{\hat{N}(\bar{y}, a)}{N(a)} \le 1 + \delta \tag{69}$$

**Equality of opportunity**  Similarly, we can optimize for other group concepts of fairness, such as *equality of opportunity*, which is satisfied when:

$$P(\hat{y} = 1 \mid y = 1, a = 1) = P(\hat{y} = 1 \mid y = 1, a = 0) \tag{70}$$

This can be computed by two separable threshold constraints:

$$g_a(\mathcal{D}, \hat{k}) = \begin{cases} \frac{|\mathcal{D}_{y,a}|}{N(y,a)} & \hat{k} = 1 \\ \frac{|\mathcal{D}_{y,\bar{a}}|}{N(y,\bar{a})} & \hat{k} = 0 \end{cases} \qquad g_{\bar{a}}(\mathcal{D}, \hat{k}) = \begin{cases} \frac{|\mathcal{D}_{y,a}|}{N(y,a)} & \hat{k} = 0 \\ \frac{|\mathcal{D}_{y,\bar{a}}|}{N(y,\bar{a})} & \hat{k} = 1 \end{cases} \tag{71}$$

In this equation $\mathcal{D}_{y,a}$ and $\mathcal{D}_{y,\bar{a}}$ are the number of instances with a positive outcome in the dataset for group $a$ and $\bar{a}$, and $N(y, a)$ and $N(y, \bar{a})$ represent the number of group $a$ and $\bar{a}$ that have the positive outcome in the data.

Table 8: Runtime (s) results for optimizing with a *demographic parity* constraint. Timeouts (>600s) are marked with '-'.

| Dataset | $|\mathcal{D}|$ | $|\mathcal{F}|$ | Aghaei [1] | | Jo [44] | | DPF [53] | | STreeD | |
|---|---|---|---|---|---|---|---|---|---|---|
| | | | d=2 | d=3 | d=2 | d=3 | d=2 | d=3 | d=2 | d=3 |
| Adult | 45222 | 17 | - | - | - | - | < 1 | < 1 | < 1 | 2 |
| Bank | 45211 | 46 | - | - | - | - | < 1 | 3 | < 1 | 4 |
| Com.&Crime | 1994 | 97 | - | - | - | - | < 1 | 5 | < 1 | 29 |
| COMPAS r. | 6172 | 9 | - | - | - | - | < 1 | < 1 | < 1 | < 1 |
| COMPAS v.r. | 4020 | 9 | - | - | - | - | < 1 | < 1 | < 1 | < 1 |
| Dutch | 60420 | 58 | - | - | - | - | < 1 | 6 | < 1 | 8 |
| German | 1000 | 69 | - | - | - | - | < 1 | 2 | < 1 | 62 |
| KDD | 284556 | 117 | - | - | - | - | 1 | 39 | 3 | 26 |
| OULAD | 21562 | 45 | - | - | - | - | < 1 | 3 | < 1 | 19 |
| Ricci | 118 | 4 | 3 | 44 | 2 | 13 | < 1 | < 1 | < 1 | < 1 |
| Stud. Math | 395 | 55 | - | - | 473 | - | < 1 | < 1 | < 1 | 2 |
| Stud. Port. | 649 | 55 | - | - | - | - | < 1 | < 1 | < 1 | 3 |

Table 9: Runtime (s) results for optimizing with an *equality of opportunity* constraint. Timeouts (>600s) are marked with '-'.

| Dataset | $|\mathcal{D}|$ | $|\mathcal{F}|$ | STreeD | |
|---|---|---|---|---|
| | | | $d = 2$ | $d = 3$ |
| Adult | 45222 | 17 | < 1 | 1 |
| Bank | 45211 | 46 | < 1 | 7 |
| Com.&Crime | 1994 | 97 | < 1 | 11 |
| COMPAS r. | 6172 | 9 | < 1 | < 1 |
| COMPAS v.r. | 4020 | 9 | < 1 | < 1 |
| Dutch | 60420 | 58 | < 1 | 7 |
| German | 1000 | 69 | < 1 | 50 |
| KDD | 284556 | 117 | 3 | 37 |
| OULAD | 21562 | 45 | < 1 | 19 |
| Ricci | 118 | 4 | < 1 | < 1 |
| Stud. Math | 395 | 55 | < 1 | < 1 |
| Stud. Port. | 649 | 55 | < 1 | 1 |

### F.3 Experiments

In our experiments we compare STreeD with the two MIP methods Aghaei-Fair-MIP [1] and Jo-Fair-MIP [44]; and the DP method DPF [53]. The comparison focuses on scalability.

**Setup** We follow the setup from [53], by testing each method on 12 datasets, preprocessed and binarized as described in [50]. Categorical variables are binarized using one-hot encoding. Categorical variables with twenty or more categories are removed. For each of these datasets, the task is to find an optimal decision tree of depth 2-3 with at most 1% discrimination. A timeout of 600 seconds is used. Tests are repeated five times and we report averages. For all methods, we show results for enforcing a demographic parity constraint. For STreeD, we also show runtime results for optimizing with an equality-of-opportunity constraint.

References to the original datasets can be found here [6, 17, 24, 27, 49, 62, 77].

**Results** Table 8 shows the runtime results. Both MIP methods almost always hit the time limit, except for the smallest datasets. In contrast, both DP methods find optimal decision trees for depth two in less than one second except for the KDD-Census income dataset. For depth three DPF outperforms STreeD. This is partly because of how the fairness constraint is formulated for STreeD: as two threshold constraints. For better performance, an objective more like the one presented in DPF

Table 10: Runtime (s) for maximizing accuracy, with timeouts beyond 600s denoted with '-'. Best results are shown in bold. Average of five runs. Datasets for which all methods finish within one second are left out.

| Dataset | $|\mathcal{D}|$ | $|\mathcal{F}|$ | $d = 4$ | | | $d = 5$ | | |
| --- | --- | --- | --- | --- | --- | --- | --- | --- |
| | | | DL8.5 | MurTree | STreeD | DL8.5 | MurTree | STreeD |
| Anneal | 812 | 93 | 2 | **< 1** | < 1 | 17 | **2** | 5 |
| Audiology | 216 | 148 | 4 | < 1 | **< 1** | < 1 | **< 1** | **< 1** |
| Australian-Credit | 653 | 125 | 23 | **1** | 2 | - | **21** | 37 |
| Breast-Wisconsin | 683 | 120 | 4 | **< 1** | 1 | 15 | 1 | 7 |
| Diabetes | 768 | 112 | 24 | **1** | 2 | - | **22** | 34 |
| Fico | 10459 | 17 | **< 1** | < 1 | < 1 | **< 1** | 1 | 2 |
| German-Credit | 1000 | 112 | 36 | **2** | 3 | - | 61 | 94 |
| Heart-Cleveland | 296 | 95 | 6 | **< 1** | < 1 | 80 | **3** | 7 |
| Hepatitis | 137 | 68 | < 1 | **< 1** | < 1 | 2 | < 1 | **< 1** |
| Hypothyroid | 3247 | 88 | 5 | 1 | **< 1** | 94 | 13 | **11** |
| Ionosphere | 351 | 445 | - | **69** | 152 | - | 146 | **112** |
| Kr-vs-kp | 3196 | 73 | 2 | < 1 | **< 1** | 18 | 6 | **4** |
| Letter | 20000 | 224 | 547 | 200 | **145** | - | - | - |
| Lymph | 148 | 68 | 1 | **< 1** | < 1 | < 1 | < 1 | **< 1** |
| Mushroom | 8124 | 119 | 3 | < 1 | **< 1** | 1 | < 1 | **< 1** |
| Pendigits | 7494 | 216 | 160 | 60 | 74 | - | 240 | **109** |
| Soybean | 630 | 50 | < 1 | **< 1** | < 1 | 6 | **< 1** | 1 |
| Splice-1 | 3190 | 287 | - | **148** | 156 | - | - | - |
| Tic-Tac-Toe | 958 | 27 | < 1 | **< 1** | < 1 | 2 | **< 1** | < 1 |
| Vehicle | 846 | 252 | 79 | **8** | 15 | - | 137 | 343 |
| Vote | 435 | 48 | < 1 | **< 1** | < 1 | 5 | **< 1** | 1 |
| Yeast | 1484 | 89 | 10 | **< 1** | 1 | 280 | **13** | 18 |
| Average rank | | | 3.0 | **1.29** | 1.71 | 2.89 | **1.39** | 1.72 |

could be formulated for STreeD. However, formulating and implementing a fairness constraint in STreeD is much easier than developing a specialized method such as DPF. For equality of opportunity, STreeD shows similar results as when optimizing demographic parity.

In summary, STreeD shows performance not too far off from an existing specialized DP method; allows for easy formulation of new types of constraints, and outperforms both MIP methods by a large margin.

## G   Classification Accuracy

The most studied use case for optimal decision trees is maximizing accuracy. Since STreeD generalizes previous dynamic programming approaches for optimal decision trees, here we evaluate how STreeD performs compared to two state-of-the-art dynamic programming approaches to optimal decision trees: DL8.5 and MurTree.

**DL8.5** [3, 4] is a dynamic programming approach that improves on DL8 [65, 66] by adding branch-and-bound and new caching techniques. We use their publicly available Python package.[7] Note that we compared to their latest available version, which incorporates ideas from MurTree [20].

**MurTree** [20] is also a dynamic programming approach for optimal decision trees. Like DL8.5 it incorporates branch-and-bound and caching. Other improvements are a similarity lower bound and a special depth-two solver. We use the publicly available Python package.[8]

---

[7]https://github.com/aia-uclouvain/pydl8.5
[8]https://github.com/MurTree/pymurtree

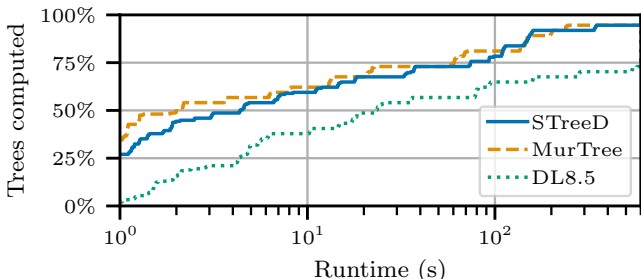

Figure 6: Runtime (s) comparison. A timeout of 300 seconds is used. Note the logarithmic x-axis.

We test both methods and STreeD on a set of binarized datasets with binary class as described in Table 10. These are the same datasets as used in Appendix E.[9] For each dataset, all three algorithms need to find a tree of depth four or five with minimum misclassification score. We repeat the tests five times and report the average runtime in Table 10. Runtime results are also shown in Fig. 6.

By using the geometric mean of the relative performance [31], we can see that MurTree on average is 1.25 times faster than STreeD and STreeD on average is 6.4 times faster than DL8.5. According to a Wilcoxon signed rank test with a significance level 5%, both results are significant.

Since STreeD outperforms DL8.5 and DL8.5 outperforms DL8, we can conclude that STreeD also outperforms DL8 in terms of scalability. Furthermore, Demirović et al. [20] compared MurTree with GOSDT [52] and observed that GOSDT resulted in a timeout for 65% of the 68 datasets when computing optimal trees with maximum depth four, whereas MurTree did not result in a single timeout. Since STreeD's performance is only a factor of 1.25 slower than MurTree, we conclude that STreeD also outperforms GOSDT.

---

[9]`https://bitbucket.org/EmirD/MurTree-bi-objective`

