# OpenReview forum: "Necessary and Sufficient Conditions for Optimal Decision Trees using Dynamic Programming"
_NeurIPS.cc/2023/Conference — NeurIPS 2023 poster_

### Official Review · Reviewer_khZh · 2023-06-18

**Soundness:** 2 fair
**Presentation:** 3 good
**Contribution:** 3 good
**Rating:** 5
**Confidence:** 4

**Summary:**

The authors present a dynamic programming (DP) method for constructing optimal decision trees.
The method is more general than previous DP methods. The authors report extensive experiments
to compare with previous methods which including MIPS and DP methods.

**Strengths:**

The paper is well-written and there is extensive supplementary material including code.
There appears to be a real contribution in the integration of diverse types of constraints
within the DP framework. The DP method is conclusively shown to be much faster than MIPS methods.
There is a welcome theoretical analysis of the problem which allows the authors to state weaker
sufficient conditions (than were previously known) for the DP method to work.

**Weaknesses:**

My main (and only major) criticism of this paper is that the authors try to prove that their list
of conditions are not only sufficient but also necessary for the DP method to work.
I believe that these conditions are sufficient, but I have some doubts about necessity.

Firstly, in lines 193-194 you give a definition of a Markovian cost function:
it is a cost function that depends just on the state and the current branching decision.
In Appendix A (lines 46-47) you give the impression that a Markovian cost function can depend on
the 'history' of branching decisions, which seems to go against standard definitions (including your own)
of Markovian. Would the DP method still work if the cost depended, for example, on the order of the
branching decisions in the history? If so, then the claim that having a Markovian cost function is
a necessary condition for the DP method to work appears false.

Secondly, you give a definition of anti-monotonicity (Def. 4.5) which appears non-standard.
I would expect c to be anti-monotonic iff (c(Y) and X a part of Y implies c(X)) but you say
c is anti-monotonic iff (opt(Y) and X a part of Y implies c(X)). You could call this, for example,
anti-monotonicity on optimal solutions. A proof of necessity can only work with this weaker version
but Appendix A (lines 114-116) seems to be saying that the stronger version is necessary (which is wrong).

One obvious fix would be to weaken your claims by dropping the 'only if' parts of Prop. 4.3
and Theorem 4.6. This would leave a very solid paper. The other solution is to try and produce
a watertight proof of the 'only if' directions in the rebuttal.

There are also a few minor presentation improvements that can be made in terms of presentation:
- In last sentence of the abstract you should say something about the quality of the trees you find
(otherwise being faster is a vacuous piece of information).
- lines 166-170: This paragraph is confusing. One possible interpretation is that constraints
are never taken into account, which is surely not the case. It is not clear what 'building blocks' refers to.
- Definition 4.4 (line 209): you should avoid phrases such as 'and thus' in a definition.
This is a consequence of the definition not part of it. This could be stated after the definition.
- Definition 4.5: again the 'and thus v_1 \notin opt(\Theta_1,s_1)' is out of place in a definition.
It is implicit that Theta/Theta_i are the set of feasible solutions for s/s_i. It would be better to recall
within the definition.
- line 218: give an example of a non-anti-monotonic constraint.
- line 229: why 'street' instead of 'streed'?
- equation (10): 'else' should be 'otherwise'
- line 285: do you mean monotonic or anti-monotonic?

**Questions:**

(1) Is the 'only if' part of Prop. 4.3 correct with the standard definition of 'Markovian'?

(2) Is there a typo in lines 114-116 of Appendix A?

(3) When you say in line 266 'This Pareto front can then be used to find the decision tree with, e.g.,
the best F1-score', is the Pareto front of possibly exponential size?

(4) In lines 268-282 (Group fairness): Are the actual constraints you apply
stronger than demographic parity. Are you imposing that you have parity down each branch, which would be much stronger?

****ADDED AFTER REBUTTAL****
I appreciated the authors' rebuttal and replies to my comments. I am generally positive about this paper, but it is important that the authors make perfectly clear to the reader what the necessary and sufficient conditions are. I found the terms Markovian and anti-monotonicity confusing and the authors should consider using history-dependent Markovian and anit-monotonicity on optimal solutions, for example.

**Limitations:**

There are no obvious negative societal impacts.

---

> ### Author Rebuttal · Authors · 2023-08-09
>
> Thank you for your positive words about our work! In response to your questions and comments:
>
> **Markovian**
>
> 1. "In Appendix A (lines 46-47) you give the impression that a Markovian cost function can depend on the 'history' of branching decisions, which seems to go against standard definitions (including your own) of Markovian"
>
> The parents’ branching decisions can be included in the state, as explained in line 163-164. This is a common trick applied in MDPs. Such an MDP is called a _history dependent_ Markov Decision Process, which is equivalent to a normal Markov Decision Process (see, for example, Section 1.4.1 in Sigaud and Buffet, _Markov decision processes in artificial intelligence_, John Wiley & Sons, 2013).
>
> 2. "Would the DP method still work if the cost depended, for example, on the order of the branching decisions in the history? If so, then the claim that having a Markovian cost function is a necessary condition for the DP method to work appears false"
>
> Yes, the DP method would still work. The order of the parents’ branching decisions could be recorded in the state. Given our answer at point 1, this does not violate our claim that a Markovian cost function is a necessary condition.
>
> 3. (Q1) "Is the 'only if' part of Prop. 4.3 correct with the standard definition of 'Markovian'?"
>
> Points 1 and 2 show that the only if part of Prop 4.3 is correct and in accordance with the standard definition of Markovian.
>
> **Anti-monotonicity**
>
> 4. "The definition of anti-monotonicity is non-standard."
>
> We indeed only require anti-monotonicity on optimal solutions, in order to have a necessary condition. Anti-monotonicity on all solutions is sufficient, but not necessary. We will update the text to make this more clear.
>
> 5. "A proof of necessity can only work with this weaker version but Appendix A (lines 114-116) seems to be saying that the stronger version is necessary (which is wrong)."
>
> Appendix A (lines 114-116) uses the weaker version, and not the stronger version. There is no typo in these lines (Q2), but we understand the confusion of our notation. We will update the text to make this more clear by replacing $\theta$ with $opt(\Theta)$ and $\theta_1$ with $opt(\Theta_1)$ in lines 114-116 in the appendix.
>
> 6. "line 285: do you mean monotonic or anti-monotonic?"
>
> This should indeed be anti-monotonic. Thanks!
>
> **Pareto front**
>
> 7. (Q3) "When you say in line 266 'This Pareto front can then be used to find the decision tree with, e.g., the best F1-score', is the Pareto front of possibly exponential size?"
>
> Let $N_0$ and $N_1$ be the number of negative and positive instances in a dataset. The maximum size of the Pareto-front is $M = \min(N_0, N_1)$. This Pareto-front could for example have the values $\\{ (M-x, x) ~ | ~ x \in \\{0, 1, ..., M\\} \\}$.
>
> **Demographic Parity**
>
> 8. (Q4) "In lines 268-282 (Group fairness): Are the actual constraints you apply stronger than demographic parity. Are you imposing that you have parity down each branch, which would be much stronger?"
>
> We enforce the demographic parity on the whole tree only, and not on each branch individually.
>
> Enforcing demographic parity on every branch would be an example of a non-anti-monotonic constraint: a subtree that exceeds a discrimination limit could be balanced out by another subtree with an opposite bias, yielding a combined tree that does not violate the demographic parity constraint.
>
> **Pronunciation**
>
> 9. "Why 'street' instead of 'streed'?"
>
> This is a play on words, since "t" and "d" are often pronounced the same (incorrectly).
>
> **Other suggestions**
>
> We also thank the reviewer for the other suggestions for improving the text. We will include the suggestions in our final version.

---

> > ### Comment · Reviewer_khZh · 2023-08-19
> > **Reply to the authors' rebuttal**
> >
> > Dear authors,
> >
> > Thank you for your detailed rebuttal.
> >
> > The rebuttal makes sense and all my questions have been answered.
> >
> > However, I am a little concerned that you have to make a change to an important claim in the paper (anti-monotonicity on optimal solutions not on all solutions) which it will not be possible to verify by another review.

---

> > > ### Author Response · Authors · 2023-08-21
> > >
> > > The changes we will make to our final text concerning anti-monotonicity will not change any important claim in our paper. These changes are only small changes to improve clarity and avoid misunderstanding, namely:
> > > 1. Addition of one sentence in the main text where we highlight that our definition of anti-monotonicity only requires anti-monotonicity of optimal solutions and that requiring all solutions to satisfy anti-monotonicity, as Nijssen and Fromont (2010) require, is not necessary.
> > > 2. We will update the notation in Appendix line 114-116 as specified to avoid misunderstanding.
> > >
> > > Our definitions, theorems and proofs in our first version already used anti-monotonicity on optimal solutions only. Therefore, none of these changes affect our theorems, proofs and definitions. The changes also do not affect our experimental results.

---

### Official Review · Reviewer_4G5j · 2023-07-04

**Soundness:** 3 good
**Presentation:** 3 good
**Contribution:** 4 excellent
**Rating:** 7
**Confidence:** 4

**Summary:**

The paper introduces STreeD, a novel dynamic programming (DP) framework designed for learning optimal decision trees. By expanding the range of solvable objectives and constraints, STreeD offers significant advancements in decision tree optimization. The authors also offer theoretical insights to aid in determining the solvability of specific optimal decision tree problems using DP. The effectiveness of STreeD is showcased through its successful application in various tasks, such as revenue maximization under capacity constraints, group fairness, and optimization for nonlinear classification metrics.

**Strengths:**

The contributions of the paper include a new cost function that allows for more flexible optimization, a new framework for learning decision trees that can handle a wider range of objectives and constraints, and empirical results demonstrating the effectiveness of the approach on several tasks. Overall, this paper is well organized and easy to follow. Additionally, the claims made in the paper are well-supported by comprehensive experimental evaluations.

**Weaknesses:**

- Regarding section 4.4, would it be possible for you to provide an example that illustrates a situation where the optimization task is non-separable?

- Regarding Figure 2, could you kindly provide a description of the relationship between the Remaining Gap at Time-out and the number of Trees Computed? The current plot exhibits an unusual pattern where the gap appears to increase as the number of computed trees increases, which appears unexpected.

- Concerning the final cost of a tree as expressed in Equation (4), could you please provide a precise explanation of the meaning of the term g(s, b(u))?

**Questions:**

see weaknesses.

**Limitations:**

The limitations of this work are thoroughly discussed in the conclusion section. However, it is important to acknowledge that the current framework is limited to parallel splits only. In future investigations, exploring other types of splits, such as oblique splits, could be considered.

---

> ### Author Rebuttal · Authors · 2023-08-09
>
> Thank you for your positive words about our work! In response to your questions:
>
> 1. "Regarding section 4.4, would it be possible for you to provide an example that illustrates a situation where the optimization task is non-separable?"
>
> An example of a problem for which we expect no efficient separable optimization task can be formulated is the optimization decision tree policies for Markov decision processes, for which Vos and Verwer present a MIP formulation (Vos & Verwer, arXiv:2301.13185, 2023). In this problem, the optimal policy in a subtree depends on the frequency of each Markov state being reached. However, the frequency of each state being reached is dependent on the policy, which depends on all decision variables in the rest of the tree. This makes it hard to see how an optimal solution for the current subtree can be found independently from the rest of the tree.
>
> We will mention this example in the final version of our work.
>
> 2. "Regarding Figure 2, could you kindly provide a description of the relationship between the Remaining Gap at Time-out and the number of Trees Computed?"
>
> The plot is a cumulative distribution plot with on the vertical axis the percentage of problem instances which were a) solved within the given runtime (left side of the plot), or b) solved up to a certain MIP gap (right side of the plot).
>
> For example in Fig. 2b, our method finds the optimal tree for all problem instances within 20 seconds. The state-of-the-art MIP method Jo-PPG-MIP (Jo et al., 2021), finds the optimal solution for 50% of the problem instances within the time-out of 300 seconds. This means the MIP gap is 0% for 50% of the problem instances (middle of the plot). At timeout, 80% of the instances are solved with a MIP gap below 50% (right side of the plot). This means that at timeout 20% of the instances still had a remaining MIP gap higher than 50%: far from being solved.
> We will add this clarification in the paper.
>
> 3. "Concerning the final cost of a tree as expressed in Equation (4), could you please provide a precise explanation of the meaning of the term g(s, b(u))?"
>
> $b(u)$ is the branching feature of node $u$. $g(s, b(u))$ is the cost of branching on feature $b(u)$ in state $s$

---

### Official Review · Reviewer_46BC · 2023-07-07

**Soundness:** 4 excellent
**Presentation:** 4 excellent
**Contribution:** 4 excellent
**Rating:** 8
**Confidence:** 3

**Summary:**

The paper a proposes a novel framework for constructing Dynamic Programming (DP) algorithms for learning decision trees on Separable objectives. Historically DP methods are among the fastest methods that build optimal decision trees, and generatlization to a wide class of objectives is a useful contribution.
Theoretical contribution includes a rigorous definition of Separable objective, and a main theorem that defined multiple properties that must be satisfied by the objective to be separable. A mathematical formulation of the DP recursive algorithm is given, that is guaranteed to work for any separable objective.
Detailed evaluation is done on multiple datasets and objectives, some of which were previously considered in the literature, and other problems settings are novel and suitable only to the proposed framework.

**Strengths:**

- This is a strong generalization of existing DP frameworks for Optimal Decision Trees solving.
- According to the experiments, this paper may set a new state of the art in terms of being the most efficient and flexible DP solver for Optimal Decision Trees.


**Weaknesses:**

- No major or minor weaknesses.


**Questions:**

-

**Limitations:**

It would be nice to add examples of non-separable objectives to outline the scope of applicability of the proposed framework; to quantify how frequently such objectives are encountered in real-world tasks and datasets.

---

> ### Author Rebuttal · Authors · 2023-08-09
>
> Thank you for your positive words about our work!
>
> An example of a problem for which we expect no efficient separable optimization task can be formulated is the optimization decision tree policies for Markov decision processes, for which Vos and Verwer present a MIP formulation (Vos & Verwer, arXiv:2301.13185, 2023). In this problem, the optimal policy in a subtree depends on the frequency of each Markov state being reached. However, the frequency of each state being reached is dependent on the policy, which depends on all decision variables in the rest of the tree. This makes it hard to see how an optimal solution for the current subtree can be found independently from the rest of the tree.
>
> We will mention this example in the final version of our work.

---

> > ### Comment · Reviewer_46BC · 2023-08-18
> >
> > I would like to thank Authors for their response and thorough examples in the rebuttals that showcase the generality and applicability of the proposed framework, which confirms my original evaluation score.

---

### Official Review · Reviewer_VtkV · 2023-07-09

**Soundness:** 4 excellent
**Presentation:** 4 excellent
**Contribution:** 3 good
**Rating:** 7
**Confidence:** 4

**Summary:**

The paper investigates the conditions under which an optimal binary decision tree problem can be formulated as a dynamic programming (DP) problem, proposing the so-called *STreeD* framework. More specifically, the text establishes a general concept of separability for the objectives and constraints of DP-representable optimization problems. This concept requires state transitions to be order-preserving (for optimality) and anti-monotonic (for feasibility). The authors demonstrate that these properties hold for non-trivial decision tree constraints and evaluate their approach across four application domains, wherein it outperforms general-purpose solvers.

**Strengths:**

+ Very well written and rigorously formalized.
+ Framework expands the class of optimization models that can be addressed via DP in an intuitive way.

The paper contributes to the recent and growing body of work on optimization models for training decision trees. I appreciated that the paper is nicely written and formalized, proposing a more intuitive framework to verify if a binary decision tree can be more compactly written as a DP. The numerical results are also well designed and inform model choice between DP and more traditional mathematical programming methods, which is a valuable contribution. Finally, another interesting aspect is that the application to non-binary trees also seems theoretically feasible to me, as it would require a discrete action set as opposed to a binary one.


**Weaknesses:**

- The originality of the generalization is unclear.
- More details needed to justify the benchmarks in the numerical section.


My major concern is that I struggle to understand the novelty of the work and its relationship to more fundamental DP theory. I believe this is an issue of presentation and framing of the work, which attempts to be somewhat broad in Section 4.

More precisely, any discrete optimization problem can conceptually be represented as a DP model given sufficient information to encode within a state. The effectiveness of the DP is directly correlated with the space complexity of the resulting state space, since models are typically solved via value enumeration or state recursion. The paper's main contribution is to show that one can leverage a more compact state representation (the dataset-depth pair) for many classes of objectives/constraints, i.e., that no additional state variables are required to enforce constraints or optimality.

However, these are fundamental questions of DP representability, and my understanding is that the paper could possibly be reinterpreting existing classical results in the area. For example, the notion of order-preserving and anti-monotonicity seems to be quite close to the concepts of monotonicity and $\thicksim$-congruence of the seminal work by Karp & Held,

Karp, Richard M., and Michael Held. "Finite-state processes and dynamic programming." SIAM Journal on Applied Mathematics 15.3 (1967): 693-718.

and references therein. The multiobjective-representable DP concept is also classical, e.g.,

Li, Duan, and Yacov Y. Haimes. "Extension of dynamic programming to nonseparable dynamic optimization problems." Computers & Mathematics with Applications 21.11-12 (1991): 51-56.

and the idea of the *merge* also shares some similarities with the concatenation concept by

Elmaghraby, Salah E. "The concept of “state” in discrete dynamic programming." Journal of Mathematical Analysis and Applications 29.3 (1970): 523-557.

Other related works include:
- Smith, Douglas R. Representation of discrete optimization problems by discrete dynamic programs. Naval Postgraduate School, 1980.
- Pollock, Stephen M., and Robert L. Smith. A formalism for dynamic programming. 1985.

Many of the conditions discussed in the works above establish when the optimal policy will be optimal and feasible, in that any state is actually capturing all the necessary information of all paths that reach it; that is, the "merge" deriving from the state transitions are sound.

My understanding is that this paper could possibly be offering a more intuitive way of checking whether these conditions hold for the special case of binary decision trees (in contrast to building more complex automata, for example).  However, the paper lacks this discussion, which I believe is important because I am not sure if order-preserving is novel, or whether it can be derived from the classical body of work above.

*Other notes*
- It is not clear from the text if the benchmark methods correspond to the state of the art; e.g., there are nonlinear formulations (such as CP) that could also be used for training.
- The paper is very well written, but in my opinion, the term "pushing the limits of DP" is not appropriate because the paper does not exhaust all possible DP-based methodologies for these problems. I would suggest something along the lines of "Dynamic Programming Representability of Optimal Decision Trees" to highlight the theoretical contributions of the paper.




**Questions:**

It would be great if authors could comment on the relationship between Proposition 4.3 and Theorem 4.6, and existing works mentioned above.

**Limitations:**

No limitations were discussed.

---

> ### Author Rebuttal · Authors · 2023-08-09
>
> Thank you for your positive words about our work! In response to your questions:
>
> **Novelty**
>
> 1. "My major concern is that I struggle to understand the novelty of the work and its relationship to more fundamental DP theory."
>
> We provide a tailored DP theory for decision trees. This has several benefits.
>
> - We draw the deep connection between decision trees and DP. It seems that the community is not aware of this connection: papers are published at prime venues on specific problems that naturally fit within our framework, e.g., nonlinear metrics (Demirović & Stuckey, AAAI-21 ); group fairness (Van der Linden et al., NeurIPS-22) and regression (Zhang et al., AAAI-23).
>
> - Given our specific setting, we show how order preservation is a necessary condition to satisfy the classic DP principle of optimality (Bellman, 1957). We also show that additivity, as required in previous general DP methods for optimal decision trees (Nijssen and Fromont, DM&KD-10) and Lin et al., ICML-20), is not necessary.
>
> - As a consequence, this also allows us to exploit tailored algorithms for decision trees, namely a specialized algorithm for computing trees of depth-two, caching subtrees, and bounds. This third point is a minor point conceptually (we include it in the Appendix) but in practice it gives significant benefits.
>
> - Lastly we also provide code for our framework which further supports the adoption of our general DP approach in the community.
>
> In regards to specific questions about general DP theory in relation to the papers mentioned:
>
> - The _monotonicity_ condition from Karp et al. indeed has similarities with our _order-preservation_ condition. Both our work and Karp et al. point out the similarity with the principle of optimality, as stated by Bellman (1957). However, a difference is that Karp et al.’s definition assumes costs to be real, completely ordered, and additive, whereas we prove that in the context of finding optimal decision trees these assumptions are not needed.
>
> - The _right congruence_ notion discussed in Karp et al. relates to identifying equivalent states, which in DP relates to identifying equivalent subproblems, but is orthogonal to anti-monotonicity and order preservation. Similar to previous DP approaches, our method caches and reuses cached solutions for equivalent subproblems.
>
> - The _multi-objective_ notation from Li et al. has similarities with our proposition (A.4) about combining separable optimization tasks into a new separable optimization task. However, Li et al. also restrict their analysis to solutions in $R^n$. The solution value for each objective is still assumed to be real, completely ordered, and additive, which limits their theory to element-wise additive optimization tasks. Our theory does not share these limitations, which we hope to exploit in future work.
>
> - Our _merge_ operation has similarities with _output concatenation_ by Elmaghraby. However, Elmaghraby assumes that the output space (solution value space) is completely ordered, whereas our work does not make this assumption.
>
> - Smith and Douglas (1980) and Pollock and Smith (1985) both  also assume a real-valued completely ordered solution value space.
>
> To strengthen our theoretical contribution we will mention these similarities and extensions on existing theory in our final version.
>
> 2. "My understanding is that this paper could possibly be offering a more intuitive way of checking whether these conditions hold for the special case of binary decision trees."
>
> Yes, we agree: one of the contributions of our work indeed is to make DP more accessible for optimizing decision trees. It also corrects previous work that limited DP for optimal decision trees to only additive optimization tasks (Nijssen and Fromont, DM&KD-10) and Lin et al., ICML-20).
>
> **Paper title**
>
> 3. "The paper is very well written, but in my opinion, the term "pushing the limits of DP" is not appropriate because the paper does not exhaust all possible DP-based methodologies for these problems. I would suggest something along the lines of "Dynamic Programming Representability of Optimal Decision Trees" to highlight the theoretical contributions of the paper"
>
> We will adopt the suggested title to highlight, as suggested, the theoretical contributions of the paper.
>
> **Benchmarks**
>
> 4. "More details needed to justify the benchmarks in the numerical section"
>
> We are surprised by this comment since we invested a considerable effort in the experimental section. We evaluated our approach on four diverse applications, surveyed related work for each application and compared to the state-of-the-art methods for each application (including heuristic and optimal methods) on many datasets. We took extra care to make sure the experiments are detailed and have a clear outcome. Note that the main points are summarized in the paper, but more details are provided in the Appendix (we will emphasize this in the paper). We would be happy to expand the experimental part should the reviewer have further suggestions.
>
> **CP and other methods**
>
> 5. "It is not clear from the text if the benchmark methods correspond to the state of the art; e.g., there are nonlinear formulations (such as CP) that could also be used for training"
>
> We surveyed the literature and compared to the state-of-the-art optimal methods for the application domains considered, except for cost-sensitive classification, where for lack of an open source optimal method, we compare with a state-of-the-art method without optimality guarantees. We can clarify this in the text.
>
> As for CP specifically, as far as we know, CP formulations for optimal decision trees have only been proposed for maximizing accuracy and not for the optimization tasks considered in our work. For maximizing accuracy, Aglin et al. (AAAI-20) already showed that the DP approach scales better than the state-of-the-art CP solution.

---

> > ### Comment · Reviewer_VtkV · 2023-08-13
> >
> > Thank you for carefully answering my questions and outlining the connection with previous works - I believe this helps in broadening the impact of the methodology. I also appreciate the clarification of the numerical results. My concern here is why those specific four benchmarks were chosen (not sure what was the measure of diversity).
> >
> > I have updated my score accordingly.

---

> > > ### Author Response · Authors · 2023-08-15
> > >
> > > Thank you once again for your comments, we agree that adding the discussion to the paper will indeed deepen our paper.
> > >
> > > Regarding the benchmark selection: our aim was to select a diverse set of benchmarks such that it is not easy to trivially adapt the algorithm of one of the applications for the other. There is no formal measure of diversity, but the problem formulations and the applications are intuitively very different. Note that typically, decision tree papers only consider one or two similar benchmarks, whereas we demonstrated the generality of our method on four benchmarks

---

### Official Review · Reviewer_L3H7 · 2023-07-09

**Soundness:** 3 good
**Presentation:** 2 fair
**Contribution:** 2 fair
**Rating:** 6
**Confidence:** 3

**Summary:**

This paper discusses the necessary and sufficient conditions for training optimal classification trees using dynamic programming (DP). In particular, the authors replace the commonly used -- and suficient -- notion of additivity by order preservation, which is shown to be necessary. The authors also present a generalized framework for modeling optimal classification trees using DP, which presents better results in some benchmarks.

*****

Following the rebuttal by the authors, I am updating my score accordingly.

**Strengths:**

The authors present a very comprehensive review of existing work and initial explanation of the ideas. I was not aware of some of the references used, such as Verwer and Zhang's proposing optimal training with MIP at the same time as Berstimas & Dunn.

Along those lines, the authors make it clear why training with DP is beneficial, as well as how much was already done in prior work.

**Weaknesses:**

I have a hard time grasping the meaning of the main result in the paper, Theorem 4.6. This is not so much about its correctness, but rather about the feeling that the definitions preceding it seem reverse-engineered to ensure that they are both necessary and sufficient. For example, there is no example helping the reader understand what order preservation means and how it is more general than additivity. If order preservation is the key element in this paper, my lack of understanding about it makes me unsure about the relevance of the main contribution.

Along the same lines, I would have appreciated a real example and discussion of anti-monotonicity, even if it is a concept already used in other papers. I would have expected that to occur with the examples in Section 4.4, but they are very briefly explained - and in cases such as prescriptive policy generation not explained at all. Moreover, DP has already been previously used to obtain optimal classification trees in all of the cases studied. And if so, what is it that we gain from the more generalized setting described in this paper?


**Questions:**

1) Can you please describe the intuition for order preservation and where it would be useful whereas additivity would not?

2) Can you please explain how anti-monotonicity relates to the applications considered?

3) Can you please describe an application that can be addressed by your setting that was not possible previously?

4) It is not clear to me how the STreeD framework benefits from the setting described in this paper to such a point that it outperforms other methods. In your opinion, what makes the setting considered in your work also more convenient computationally?

5) Do you see a possible application of this or a related setting to optimally train decision diagrams for classification?

In terms of notation, I would caution the authors about the use of $\mathcal{D}$ to sometimes represent the entire dataset (as it seems implied in Lines 112 and 165) and sometimes represent a subset of the dataset (such as in the recurrence in equation (2)).

**Limitations:**

I do not recall seeing a discussion about limitations, but it would be fair to say that outlining what optimal classification trees can and cannot be trained with DP represents an important study about the limitations of a particular form of training algorithm.

---

> ### Author Rebuttal · Authors · 2023-08-09
>
> Thanks for the review and for the positive words about our work! We here respond to each of your questions:
>
> **Novelty**
> 1. "What is it that we gain from the more generalized setting described in this paper?"
>
> Recent publications at premier venues on using DP for optimal DTs for varying optimization tasks show that there is a great interest in this topic. E.g.:
> Accuracy (Aglin et al., AAAI-20);
> Nonlinear metrics (Demirović & Stuckey, AAAI-21);
> Group fairness (Van der Linden et al., NeurIPS-22); and
> Regression (Zhang et al., AAAI-23).
>
> We provide a general framework that covers all these cases, including regression. Normally, for each of these a specialized method had to be developed. We also precisely characterize necessary and sufficient conditions for the use of DP for new optimization tasks.
>
> This adds to our understanding of decision tree algorithms and allows us to quickly model solutions to new optimization tasks, such as for example, individual fairness, for which currently no optimal DP method exists (see next answer).
>
> 2. (Q3) "Can you please describe an application that can be addressed by your setting that was not possible previously?"
>
> Consider individual fairness as an example. Aghaei et al. (AAAI-19) have proposed a MIP method for individual fairness, but currently no optimal DP solution for individual fairness exists yet. However, this is possible within our framework. We provide details at the end of this response.
>
> **Order preservation**
>
> 3. (Q1) "Can you please describe the intuition for order preservation and where it would be useful whereas additivity would not?"
>
> Recent work showed that DP could also be used for problems that are not additive, e.g., nonlinear metrics (Demirović and Stuckey, AAAI-21) and group fairness (Van der Linden et al., NeurIPS-22). This motivates the search for the limits of the use of DP for optimizing decision trees, which we provide.
>
> The intuition behind order preservation is the principle of optimality (Bellman, 1957): optimal solutions can only be constructed from optimal solutions to subproblems.
>
> **Anti-monotonicity**
>
> 4. (Q2) "Can you please explain how anti-monotonicity relates to the applications considered?"
>
> We model group fairness as an anti-monotonic constraint. If it can be proven for a subtree that it could never be part of a tree that satisfies demographic parity, this subtree can be discarded.
>
> **Performance**
>
> 5. (Q4) "It is not clear to me how the STreeD framework benefits from the setting described in this paper to such a point that it outperforms other methods. In your opinion, what makes the setting considered in your work also more convenient computationally?"
>
> We employ dynamic programming, which exploits the fact that subtrees can be solved independently (if the conditions we present hold) and repeated subproblems can be cached. Other methods, such as MIP, do not consider this, so these methods end up doing exponentially more work, which is reflected in our experiments.
>
> The disadvantage of DP is that it is specific to each possible application. The added value of the STreeD framework compared to existing DP approaches addresses exactly this: it provides a general approach to using DP for building optimal decision trees.
>
> **Decision Diagrams**
>
> 6. (Q5) "Do you see a possible application of this or a related setting to optimally train decision diagrams for classification?"
>
> Decision diagrams are not separable in the same way as decision trees: i.e., subdiagrams may share nodes, whereas subtrees never share nodes. Therefore, the same breakdown into independent subproblems cannot trivially be applied and the question whether one could devise a DP algorithm that is more efficient for decision diagrams, remains an open question.
>
>
> **Appendix: Individual fairness separable formulation**
>
> Individual fairness optimizes the number of similar individuals (as defined by some distance function) that receive the same label.
> The following provides the details of how individual fairness could be modeled as a separable optimization task.
>
> Let $d(x_1, x_2)$ be a distance function that returns one if $x_1$ and $x_2$ are similar, and zero otherwise. Let $O(D) = \\{ (x_1, x_2) \in D | d(x_1, x_2) = 1 \\}$. Let $n = |O(D)|$ over the original dataset $D$.
>
> Whenever a node is split, the transition function should update the state $s$ to record which pairs in $O(D)$ end up in different subtrees. For the subtree with dataset $D’$, call this record $M = \\{ (x_1, x_2) \in O(D) | (x_1 \in D’ \wedge x_2 \notin D’) \vee (x_1 \notin D’ \wedge x_2 \in D’) \\}$.
>
> A solution value consists of bounds on the worst and best case value for the individual fairness, and a label for each pair in $M$: $(worst, best, L: M \rightarrow K)$. This is computed as follows: $g(D, M, \hat{k}) = (|O(D)| / n, 1, L(m) = \hat{k},  \forall m \in M)$.
> The worst case is lower bounded by $|O(D)|/n$ because all pairs in $O(D)$ receive the same label. The best case is still 1, in case all instances in $M$ receive the same label. The label of all pairs in $M$ are set to $\hat{k}$.
> When we merge two solution values $(worst_1, best_1, L_1)$ and $(worst_2, best_2, L_2)$ for solutions generated for state $(D_1, M_1)$ and state $(D_2, M_2)$, we check which split pairs in $O(D)$ are joined again: $J = \\{ M_1 \cap M_2 \\}$. Let $v = |\\{m \in J|L_1(m) = L_2(m)\\}|$ be the number of pairs with the same label. The merged solution value becomes:
> $(worst_1 + worst_2 + (|J| - v)/n, 1 - ((1-best_1) + (1-best_2) + v/n)$, $L(m) = L_1(m) \text{ if } m \in M_1, \text{otherwise } L_2(m)~\forall m \in M )$.
>
> A solution is dominated if its worst fairness value is higher than another solution’s best fairness value.
> A solution is infeasible if its worst fairness value is higher than a predetermined threshold.
>
> The optimization task described above is separable, but not additive. it satisfies all conditions of our framework, and thus results in optimal solutions.

---

> > ### Comment · Reviewer_L3H7 · 2023-08-13
> >
> > I appreciate the comments by the authors and have updated my score accordingly.
> >
> > I agree with the points raised by reviewer VtkV about the scope and significance of the work, and I second that reviewer's suggestion of a more specific and meaningful title for this paper. I am counting on the word of the authors about changing it.

---

### Comment · Area_Chair_bWxa · 2023-08-18

I would like to thank the authors for providing detailed responses to all referee reports, and I apologize that some of the referees have not responded to the rebuttals despite multiple reminders. I will account for that in my final recommendation.

---

### Decision · Program_Chairs · 2023-09-21

**Decision:**

Accept (poster)

**Comment:**

I am happy to propose acceptance of the paper to NeurIPS.

Two of the reviewers suggested a change in the title, which I would like to encourage the authors to seriously consider as well. All other comments of the review team have been answered convincingly; in those places where the reviewers have not confirmed that themselves, I checked myself that the responses resolve any of the raised issues.

Thank you for submitting your work to NeurIPS!